# Single nucleosome imaging reveals principles of transient multiscale chromatin reorganization triggered by histone ADP-ribosylation at DNA lesions

Fabiola García Fernández[1], Junwoo Park [1], Catherine Chapuis [2], Eva Pinto Jurado[2,3,4], Victor Imburchia[2], Rebecca Smith[2], Edoardo José Longarini[4,9], Angela Taddei [5], Christian Hubert[6], Nataliya Sokolovska[1], Ivan Matić [7,8], Sébastien Huet [2] ✉ & Judith Miné-Hattab[1] ✉

Timely access to DNA lesions is crucial for genome integrity. This process requires profound remodeling of densely packed chromatin to establish a repair-competent architecture. However, limited resolution has made it impossible to fully understand these remodeling events. Here, combining microirradiation with live-cell multiscale imaging, we report that DNA damage-induced changes in genome packing rely on the conformational behaviour of the chromatin fiber. Immediately after damage, a transient increase in nucleosome mobility switches chromatin from a densely-packed state to a looser conformation, making it accessible to repair. While histone poly-ADP-ribosylation is required to trigger this switch, mono-ADP-ribosylation is sufficient to maintain the open-chromatin state. The removal of these histone marks by the ARH3 hydrolase then leads to chromatin recondensation. Together, our multiscale study of chromatin dynamics establishes a global model: distinct waves of histone ADP-ribosylation control nucleosome mobility, triggering a transient breathing of chromatin, crucial for initiating the DNA damage response.

The high level of DNA packing displayed by chromatin in the cell nucleus represents a major challenge for DNA-transaction processes, including the repair of genetic alterations. The DNA damage response (DDR) is characterized by multiple chromatin remodeling processes.

Among them, histone tails undergo post-translational modifications[1], facilitating efficient and faithful genomic restoration. One of the earliest remodeling events is the rapid and transient relaxation of the chromatin architecture occurring within seconds after DNA damage

[1]Laboratory of Computational,Quantitative and Synthetic Biology, CNRS, Institut de Biologie Paris-Seine, Sorbonne Université, Paris, France. [2]Univ Rennes, CNRS, IGDR (Institut de Génétique et Développement de Rennes)-UMR 6290, BIOSIT-UMS 3480, Rennes, France. [3]Laboratory of DNA Damage and Nuclear Dynamics, Institute of Genetics, Biological Research Centre, Eötvös Loránd Research Network (ELKH), Szeged, Hungary. [4]Doctoral School of Multidisciplinary Medical Sciences, University of Szeged, Szeged, Hungary. [5]Institut Curie, Université PSL, Sorbonne University, CNRS, Nuclear Dynamics, Paris, France. [6]Errol laser, Saint Maur des Fossés, France. [7]Research Group of Proteomics and ADP-Ribosylation Signaling, Max Planck Institute for Biology of Ageing, Cologne, Germany. [8]Cologne Excellence Cluster for Stress Responses in Ageing-Associated Diseases (CECAD), University of Cologne, Cologne, Germany. [9]Present address: Department of Chemistry, Princeton University, Princeton, NJ, USA. ✉e-mail: sebastien.huet@univ-rennes.fr; judith.mine-hattab@sorbonne-universite.fr

induction to facilitate access to the lesions[2–6]. This rapid loosening is triggered by ADP-ribosylation (ADPr), a modification known to contribute to several repair pathways, such as DNA strand breaks resolution[7]. Upon recruitment to DNA lesions, the polymerase PARP1 adds ADP-ribose on the serine residues of nearby proteins, primarily PARP1 itself and histones[8,9]. While this signaling pathway has been usually considered as mainly composed of poly-ADP-ribose (PAR) chains, recent findings evidenced a distinct, more enduring, mono-ADP-ribose (MAR) signal, potentially displaying specific functions[10,11]. The homeostasis of these two components of the ADPr pathway is controlled by HPF1, a PARP1 cofactor regulating its catalytic activity[12–14], as well as hydrolases preferentially targeting PAR or MAR marks[15–18]. While PARG is the most active PAR hydrolase, ARH3 is a specific serine MAR eraser[16]. PARP1 has been identified as a central regulator of the chromatin architecture for several decades[7]. In vitro, PARP1 binding to nucleosomes was reported to promote the compaction of isolated chromatin fiber[19] while PARP1 catalytic activity was rather involved in the decondensation of the fiber[20]. More recently, live-cell experiments have shown that PARP1-dependent histone ADPr regulates chromatin compaction state in the vicinity of DNA breaks[4,21,22]. Nevertheless, it remains unknown how this modulation of chromatin compaction relates to conformational changes at the level of the chromatin fiber. As for any polymer, there is an intimate relationship between the chromatin architecture and its dynamics, although the exact characteristics of this relationship remain only partially understood[23]. Multiple studies have reported an increase in chromatin dynamics upon DNA damage, indicative of major changes in the underlying chromatin architecture[24–33]. Such increase in chromatin dynamics is now considered as an integral part of the DDR, for example favoring homology search during homologous recombination (HR)[33–35]. However, this compelling model originates mainly from studies in yeast, leaving the actual picture in mammalian cells more ambiguous[36]. Depending on the kind of DNA damage, the distance from the lesions as well as the time after damage induction, various impacts on the local dynamics of the chromatin fiber have been reported in mammalian systems[26,27,37–39]. More importantly, previous studies have traditionally focused on a single spatial scale, preventing the establishment of a comprehensive model for changes in chromatin structure in the DDR.

In the present work, we combine single-molecule imaging and micro-irradiation in human cells to dissect the remodeling events undergone at different scales of the chromatin structure during early steps of the DDR. This multiscale approach enabled us to discover that, within the first seconds after DNA damage, a temporary increase in nucleosome mobility alters chromatin from a densely packed state to a looser conformation, making it accessible to the repair machinery. Moreover, building on the recent surge of new insights into ADPr signaling, we demonstrate that histone ADPr is a master regulator of these remodeling events, with differential roles played by the PAR and MAR signals. Our findings provide a solution to the decades-long puzzle of reconciling the transient nature of poly-ADPr with the enduring effect of PARP1 on chromatin by assigning an open chromatin maintenance function to histone mono-ADPr.

## Results

### Multi-scale chromatin remodeling occurs immediately after DNA damage

In order to get a comprehensive view of chromatin behavior immediately after DNA damage, we assessed chromatin dynamics at three folding scales: the global compaction state, the chromatin fiber and the nucleosome. To measure changes affecting the compaction state, we irradiated the nucleus of Hoechst-presensitized human U2OS cells expressing H2B fused to the photo-activatable green fluorescent protein (PAGFP) using a continuous 405 nm laser (Fig. 1A). Such irradiation simultaneously induces DNA lesions and highlights the damaged

chromatin. We monitored the thickness of the photoconverted line to assess changes in the level of chromatin compaction. In agreement with our previous findings[4], we observed a rapid chromatin relaxation at DNA damage sites peaking 1 min after damage. Noteworthy, this rapid decondensation, which is also observed upon irradiation with a pulsed 355 nm laser (Supplementary Fig. 1A), is not associated with significant nucleosome disassembly and rather relies on conformational changes undergone by the chromatin fiber[4]. Following the rapid relaxation phase, chromatin remained in an open state for a few minutes and then slowly recondensed to ultimately reach a compaction level that is beyond the pre-damage state (Fig. 1A), in line with previous findings[40]. We have previously shown that DNA gets more accessible upon chromatin decondensation at sites of damage, thus promoting the accumulation of DNA-binding sensors[5]. Here, we monitored the recruitment of the DNA-binding domain BZIP from C/EBPa at DNA lesions and found that the initial accumulation of this sensor is followed by a slow release (Supplementary Fig. 1B), closely correlating with the kinetics of the chromatin loosening and recondensation phases. This indicates that, following an initial increase upon decondensation, the accessibility to DNA gets progressively restored to its predamage level.

Next, we analyzed how these rapid changes in the compaction state correlated with a modulation of the conformational behavior of the chromatin fiber. First, we assessed the dynamics of the fiber in cells harboring a *lacO* array, inserted at a single genomic location in a euchromatic region of chromosome 1, visualized with GFP fused to LacI[41]. We monitored the dynamics of the tagged locus before, 1 and 10 min after DNA damage induced by irradiation with a pulsed 355 nm laser, nearby the locus or away from it (Fig. 1B). The motion of the locus was quantified by computing the mean square displacement (MSD) curves from 10 individual trajectories (Fig. 1B). In agreement with previous reports[42–48], this analysis revealed a subdiffusive behavior prior to DNA damage (MSD(t) $\sim 0.004\, t^{0.59}$, $R^2 = 0.994$) consistent with the Rouse model previously used to describe chromatin motion[44,49,50]. One minute after irradiation nearby the *lacO* array ($\simeq 1\,\mu m$), we observed an increase in chromatin dynamics as shown by the higher amplitude of the MSD curve (Fig. 1B, red curve). The fitting of the MSD revealed a complex diffusion behavior. While, at short time scales ($t < 5\,s$), the motion remained subdiffusive, although with a higher anomalous exponent (MSD $\sim 0.003\, t^{0.9}$, $R^2 = 0.997$), at longer time scales, the locus rather exhibited a directed motion (MSD $\sim 0.0002\, t^{1.9}$, $R^2 = 0.998$). Such directed motion at long timescale is consistent with chromatin decondensation that tends to push chromatin away from the irradiated area, as previously reported[4]. Ten minutes after damage, chromatin recovers its initial dynamics in correlation with its recompaction. In contrast to these changes occurring nearby DNA lesions, when irradiation was performed away from the fluorescent locus ($\simeq 8\,\mu m$), the MSD followed anomalous diffusion (MSD $\sim 0.006\, t^{0.5}$, $R^2 = 0.994$) similar to locus dynamics prior to DNA damage (Fig. 1B, green curve). Altogether, this analysis reveals a striking increase in the dynamics of the chromatin fiber in the vicinity of the DNA damage region.

To increase the resolution of our analysis one step further, we monitored the impact of DNA damage induction on the dynamics of individual histones. Using H2B fused to HaloTag (H2B-Halo) and labeled with the photoactivable Janelia Fluor PA-JF549 HaloTag ligand[51], we combined laser micro-irradiation and single molecule imaging to follow the 2-dimensional trajectories of individual nucleosomes (Fig. 1C, left). While almost no traces could be recovered in unlabeled control cells, we obtained thousands of tracks per nuclei expressing H2B-Halo, with a mean track length of about 17 frames (Supplementary Fig. 2A–C and Supplementary Movie 1). In line with previous reports[52–55], the tracks showed very limited motion of H2B proteins in the absence of damage, in contrast to freely diffusive HaloTag fused to a nuclear localization signal (Supplementary Fig. 2D).

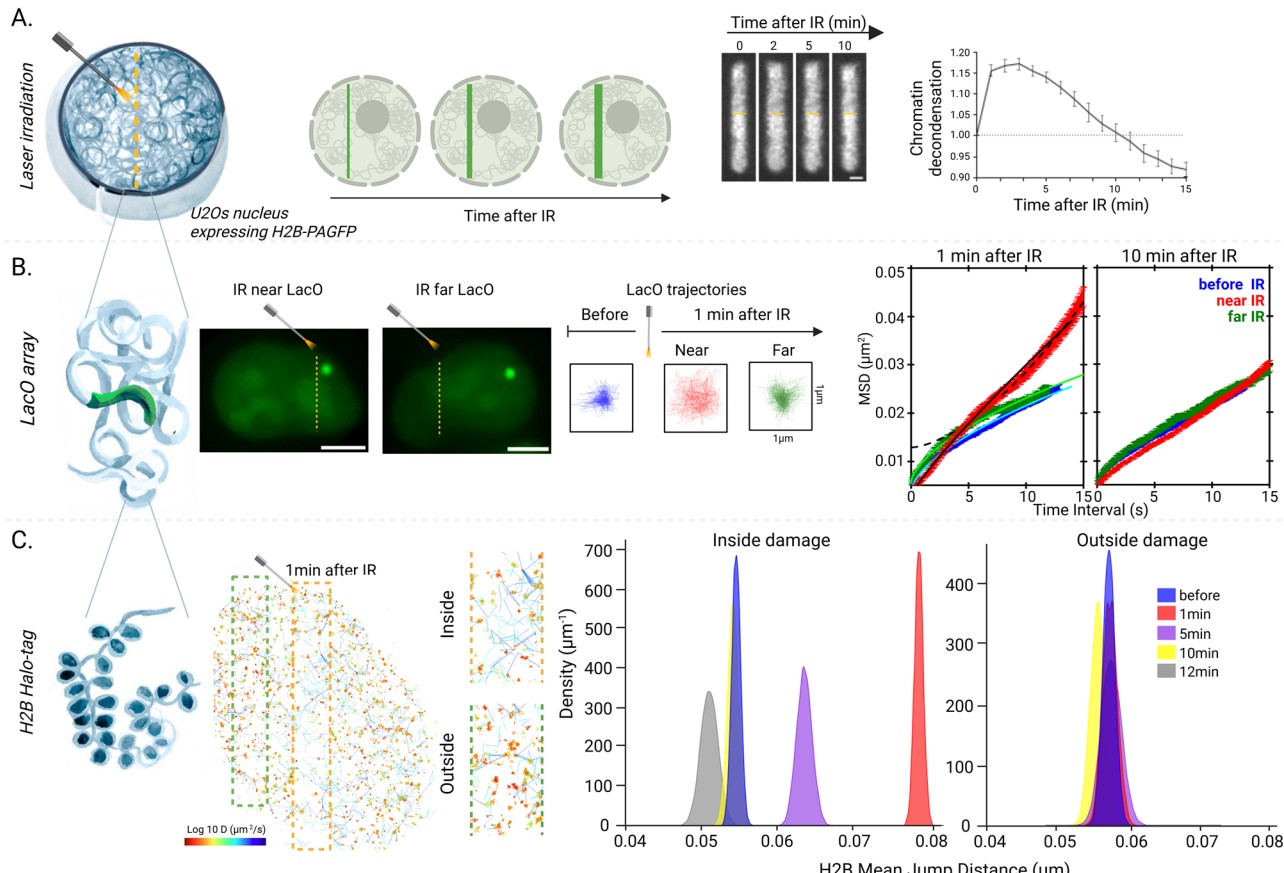

**Fig. 1 | Chromatin undergoes rapid multiscale remodeling at DNA lesions.**
**A Nuclear scale**. Sketch and representative confocal image sequence of a sub-region of a U2OS nucleus expressing H2B-PAGFP and presensitized with Hoechst after irradiation with a continuous 405 nm laser. Scale bar: 2 μm. The average thickness of the photo-activated line is plotted as a function of time after irradiation and normalized to time zero to estimate the changes in the overall chromatin compaction state ($n = 16$, mean ± SEM.). **B Chromatin fiber scale**. Representative images of U2OS nucleus harboring a fluorescent *lacO* array and irradiated with a pulsed 355 nm laser nearby or away from the array. Scale bars: 8 μm. Representative locus trajectories and MSD curves ($n = 10$) before damage (blue) and 1 min and 10 min after irradiation near (red) or far from the locus (green). Scale bar: 1 μm. Fitting of sub-diffusive and directed motion regimes is represented by solid and dotted curves, respectively. **C Nucleosome scale**. Trajectories of individual histones in a U2OS nucleus expressing H2B-Halo bound to PA-JF549 Halo ligand. H2B motions were monitored 1 min after irradiation at 355 nm. A zoom of the tracks inside and outside the irradiated area is shown on the right. The trajectories are color-coded according to their diffusion coefficient using the look-up table shown below. Mean jump distance kernel density estimation (KDE) plots for the immobile H2B tracks inside (left) and outside (right) the irradiated region, before and at different times after micro-irradiation. Total number of cells analyzed in four independent experiments ($N$). Inside damage: $N_{bef} = 44$, 1558 trajectories, $N_{1min} = 25$, 1502 trajectories, $N_{5min} = 37$, 1038 trajectories, $N_{10min} = 27$, 787 trajectories, $N_{12min} = 13$, 319 trajectories; Outside damage: $N_{bef} = 20$, 676 trajectories, $N_{1min} = 20$, 494 trajectories, $N_{5min} = 11$, 301 trajectories, $N_{10min} = 11$, 488 trajectories. The 95% confidence intervals of bootstrapped mean jump-distance inside IR are [0.0617 μm, 0.0635 μm], [0.0773 μm, 0.0795 μm], [0.0614 μm, 0.0654 μm], [0.0553 μm, 0.0605 μm], and [0.0542 μm, 0.0572 μm] for each condition. Mean jump distances between each condition *versus* before damage are significantly different inside the irradiated region ($p < 0.001$, two-sided Yuen–Welch test) but not outside ($p > 0.05$). Created in BioRender. Garcia Fernandez, F. (2025) https://BioRender.com/5akfy1x.

To analyze the tracks, we developed a convolutional neural network (CNN) deep-learning algorithm named Histone Trajectory Classification (HTC, see "Method" and Supplementary Fig. 2E, F). Using HTC, H2B trajectories were classified in different populations according to their diffusive properties (Supplementary Fig. 2G). The slow population, which gathers the vast majority of the tracks ($84 \pm 6\%$), most likely corresponds to H2B proteins stably incorporated into the nucleosomes as they display an effective diffusion coefficient similar to that of the chromatin fiber assessed with the *lacO* array ($D_{H2B} = 0.0078$ μm²/s and $D_{lacO} = 0.001$ μm²/s). The mobile fraction ($9 \pm 5\%$ of the tracks) shows a diffusion coefficient ($D_{H2B} = 3.661$ μm²/s) that is more than two-order of magnitude larger than the immobile population. Together with a hybrid population switching from immobile and mobile phases ($7 \pm 2\%$ of the tracks), these fast trajectories are probably associated with the small fraction of histones that are not stably associated with the chromatin fiber and therefore rapidly diffuse within the nucleoplasm.

Then, we applied our HTC analysis pipeline to study the behavior of individual H2B proteins in and out the area of DNA damage (Fig. 1C). Of note, we controlled that histone mobility was not affected by the successive imaging sequences used to monitor H2B trajectories at different timepoints after DNA damage (Supplementary Fig. 2H). We found that the proportions of the different populations of tracks were only mildly impacted by damage induction (Supplementary Fig. 2G), in line with our previous observations showing no major nucleosome disassembly at these early steps of the DDR[4]. For the rest of this study, our analysis of histone motion focused on the population of slowly moving H2B proteins likely incorporated into the nucleosomes. Histone mobility was assessed by measuring the mean distance covered in 10 ms for each H2B trajectory, a generic metric that did not require to assume a specific diffusion model. The data are then presented as the probability density plot of bootstrap distribution, calculated using a kernel density estimation (KDE) (see "Methods"). A rapid surge in mobility restricted to the irradiated region was observed, culminating

in $a \sim 50\%$ increase in nucleosome motion at 1 min post-damage (Fig. 1C). Importantly, this increased dynamics is not only observed when analyzing the histone tracks using our HTC approach, but also with the previously published vbSPT method[56] (Supplementary Fig. 2I), demonstrating the robustness of this finding. Previous reports showed that histone mobility is reduced in chromatin located at the nuclear rim in line with its heterochromatic state[46]. We confirmed this finding and observed increased dynamics at sites of damage regardless of the position of the histones within the nucleus (Supplementary Fig. 3). Therefore, histones undergo a leap in mobility upon damage in both euchromatin or heterochromatin area. This dramatic increase was only transient, with a rapid recovery as early as 5 min after irradiation, the nucleosomes becoming even less mobile than prior to damage at later timepoints (Fig. 1C). Comparing these data with the changes in the overall chromatin compaction state and DNA accessibility (Fig. 1A and Supplementary Fig. 1B) shows that the acute increase in nucleosome dynamics correlates with the rapid relaxation process and increased DNA accessibility. In contrast, chromatin remains in this decompacted state for several minutes despite the rapid drop in nucleosome dynamics. Therefore, while the increase in nucleosome mobility seems to underlie chromatin unpacking, it does not appear necessary for the maintenance of the open state.

## PARP1-mediated ADPr signaling is the central trigger of multi-scale chromatin remodeling at DNA lesions

In line with the rapid recruitment of PARP1 at sites of laser irradiation (Supplementary Fig. 4A), we previously showed that ADPr signaling controls the early modulation of chromatin compaction state at DNA lesions[4]. Here, we investigated whether these changes in chromatin packing could be linked to ADPr-dependent remodeling events at the level of the chromatin fiber. We monitored fiber and nucleosome dynamics in the presence of the clinically-relevant PARP inhibitor (PARPi) Talazoparib that impedes the catalytic activity of PARP1 and leads to its prolonged retention at DNA lesions (Supplementary Fig. 4A), in line with previous observations[57]. While PARPi treatment slightly but significantly increased the dynamics of the *lacO* array and nucleosomes in the absence of damage (Supplementary Fig. 4B, C), it led to decreased mobility in both assays after laser irradiation (Fig. 2A, B and Supplementary Fig. 4B). Therefore, PARPi did not only suppress the increased chromatin movements observed at sites of damage in untreated cells, but even reduced these movements. We also assessed chromatin dynamics in cells knocked out (KO) for PARP1, the main driver of ADPr signaling in the context of the DDR[58]. The loss of PARP1 suppressed the increased nucleosome mobility observed in wild-type (WT) cells at sites of damage, but did not lead to reduced

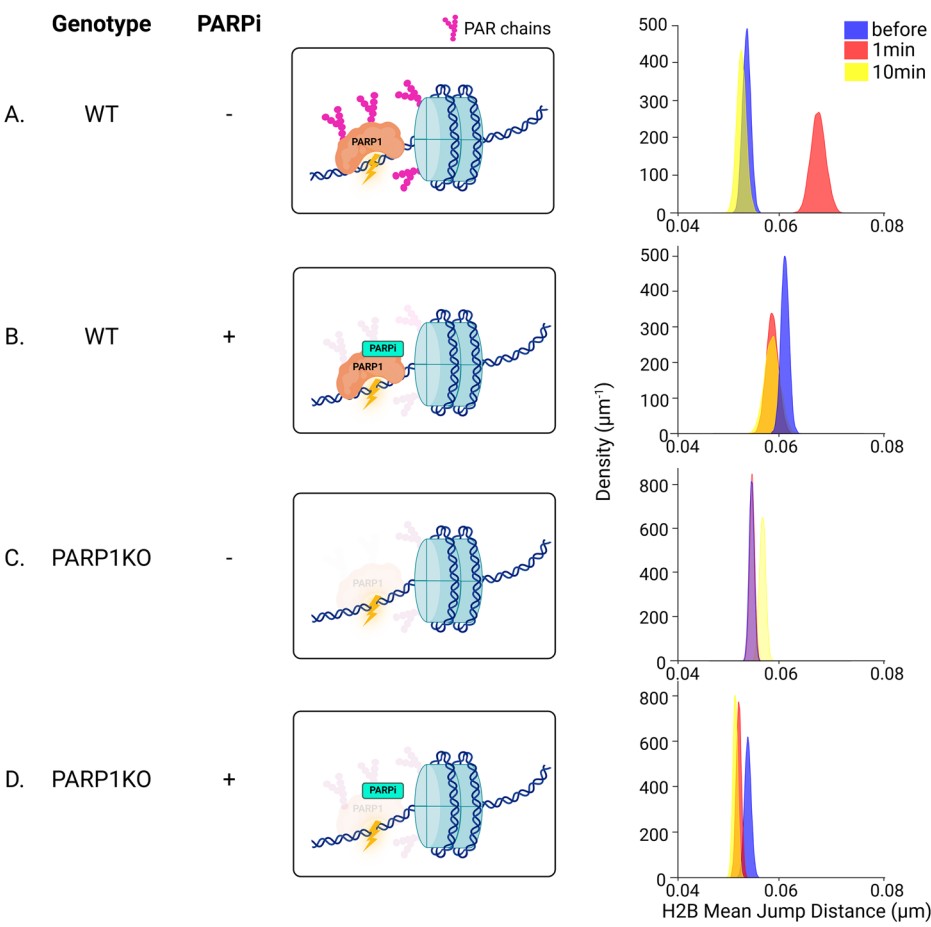

**Fig. 2 | ADP-ribosylation by PARP1 triggers enhanced nucleosome dynamics after micro-irradiation.** On the left of each panel, a sketch shows the status of the ADP-ribose signal depending on the U2OS genotype and PARPi treatment. On the right is shown the mean jump distance KDE plots for the immobile population of H2B tracks inside the irradiated region, before and at different times after micro-irradiation at 355 nm in WT (**A**, **B**) and in PARP1 KO (**C**, **D**) cells treated or not with 30 μM Talazoparib. Total number of cells analyzed in three independent experiments (N). WT $N_{bef}$ = 19, 809 trajectories, $N_{1min}$ = 12, 470 trajectories, $N_{10min}$ = 19, 490 trajectories; WT + PARPi $N_{bef}$ = 31, 868 trajectories, $N_{1min}$ = 19, 320 trajectories, $N_{10min}$ = 6, 181 trajectories; PARP1 KO $N_{bef}$ = 13, 1786 trajectories, $N_{1min}$ = 12, 1673 trajectories, $N_{10min}$ = 8, 1118 trajectories; PARP1 KO + PARPi $N_{bef}$ = 9, 1063 trajectories, $N_{1min}$ = 9, 1727 trajectories, $N_{10min}$ = 9, 1781 trajectories. Created in BioRender. Garcia Fernandez, F. (2025) https://BioRender.com/5akfy1x.

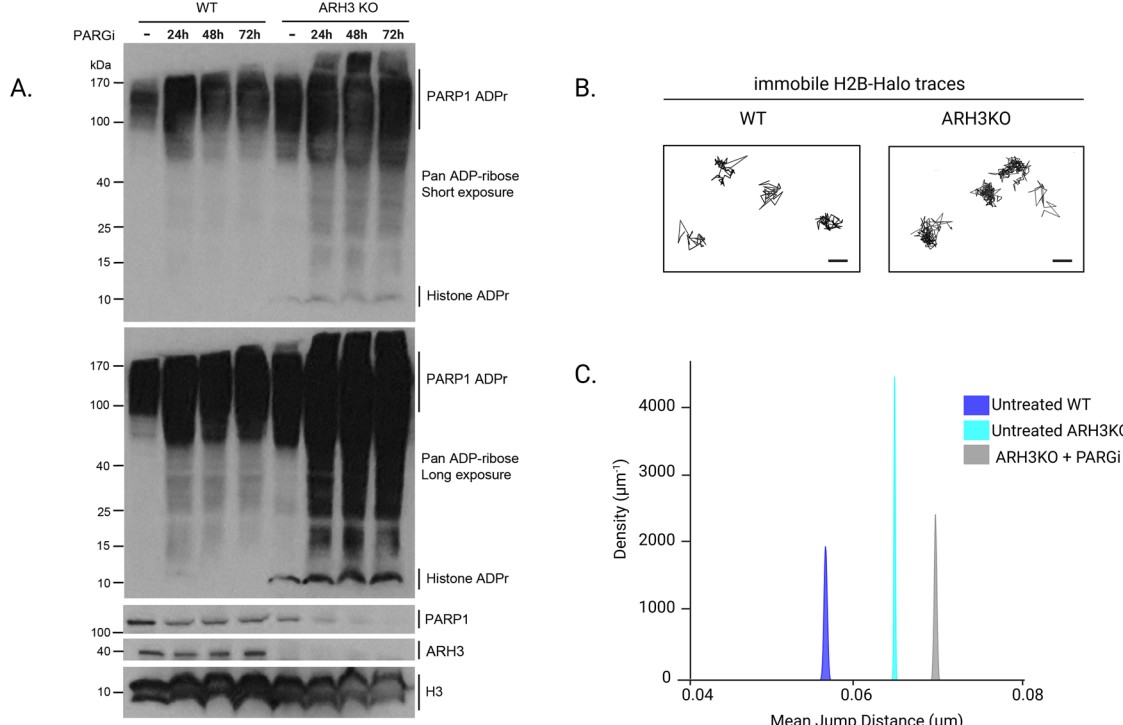

**Fig. 3 | Spontaneous ADPr signal upon loss of ARH3 is sufficient to increase nucleosome dynamics. A** Western blot displaying APDr signal, stained with a pan-ADPr antibody as well as PARP1 and ARH3 signals in WT or ARH3 KO cells, left untreated or after 24 to 72 h of PARGi treatment (25 μM PDD00017273). H3 is used as loading control (*n* = 3). **B** Representative examples of the immobile population of H2B trajectories inside the nucleus of undamaged WT and ARH3 KO cells. **C** Mean jump distance KDE plots for the immobile population of H2B tracks in undamaged

WT and ARH3 KO cells, treated or not with 25 μM PDD00017273 PARGi for 24 h. Total number of cells analyzed in at least three independent experiments (*N*). WT *N* = 35, 13769 trajectories; ARH3 KO *N* = 55, 81246 trajectories; ARH3 KO + PARGi *N* = 22, 23985 trajectories. Mean jump distance between each condition *versus* WT condition are significantly different (*p* < 0.001, two-sided Yuen–Welch test). Created in BioRender. Garcia Fernandez, F. (2025) https://BioRender.com/5akfy1x.

dynamics (Fig. 2C). Therefore, while PARP1-dependent ADPr increases nucleosome motions at sites of damage, the recruitment of inhibited PARP1 seems to restrain them. Surprisingly, in PARP1 KO cells treated with PARPi, we also observed reduced histone mobility after laser irradiation (Fig. 2D). Given that the PARPi that was used also targets PARP2, this drop might be the consequence of the accumulation of inhibited PARP2 at the breaks. Altogether, these findings nicely correlate with our previous[4] and current results (Supplementary Fig. 1A) regarding the global chromatin packing state, which showed that inactive PARP enhances chromatin compaction at DNA lesions while ADPr promotes loosening.

### Spontaneous increase in ADPr signaling is sufficient to increase chromatin fiber dynamics

Besides ADPr, the cell activates an intricate network of signaling pathways at sites of DNA damage[59]. Therefore, it is difficult to assign the changes in chromatin dynamics we observed at the lesions to a direct effect of ADPr rather than a potential crosstalk of this signaling with other DDR-related pathways. To assess the specific impact of ADPr on chromatin dynamics, we took advantage of cells lacking the hydrolase ARH3 that show spontaneous activation of ADPr signaling, in particular upon treatment with an inhibitor against the poly-ADP-ribose-glycohydrolase (PARGi) (Fig. 3A). Importantly, this was not associated with enhanced γH2AX signaling, a classical responder of DNA breaks, showing that the strong ADPr signal observed in these cells is not the consequence of a global activation of the DDR[17]. Therefore, comparing WT and ARH3 KO cells treated or not with PARGi allows for assessing the specific impact of ADPr signaling on chromatin folding independently of the DDR context. We found that nucleosome dynamics was higher in ARH3 KO compared to WT cells and could be

further enhanced by PARGi treatment (Fig. 3B, C). These data indicate a clear correlation between the level of activation of the ADPr pathway and histone dynamics, independently of the presence of DNA lesions. Therefore, enhanced ADPr appears sufficient to promote the local mobility of the nucleosomes along the chromatin fiber.

### Histone ADPr is needed to establish a dynamic open chromatin state at DNA lesions

Upon DNA damage, ADPr signal is found mainly on PARP1 and the different histones[8,18,60]. To disentangle the relative contributions of PARP1 automodification and histone ADPr on chromatin dynamics, we studied the impact of two PARP1 mutants. In the PARP1-3SA, the three main Ser residues targeted by ADPr (S499, S507, S519) are switched to Ala, leading to a strong decrease in automodification while not affecting histone ADPr[61,62]. Instead, the PARP1-LW/AA mutant (L1013A/W1014A) is unable to ADP-ribosylate histones due to impaired interaction with HPF1[12,21]. While the expression of WT PARP1 or PARP1-3SA in PARP1 KO cells both rescued the transient increase in nucleosome mobility at sites of DNA lesions, this was not the case for PARP1-LW/AA (Fig. 4). These data demonstrate that it is the ADPr of histone and not PARP1, that triggers the increase in nucleosome mobility at sites of DNA breaks. Together with the fact that histone ADPr was also shown to control chromatin relaxation at the lesions[21], our findings draw a model in which the addition of ADPr marks along the chromatin fiber increases its mobility, which itself promotes global reorganization.

### The erasure of mono-ADP-ribose is needed for chromatin recondensation

Consecutively to its initial rapid relaxation, chromatin remained in an open state for a few minutes. This was followed by a slow

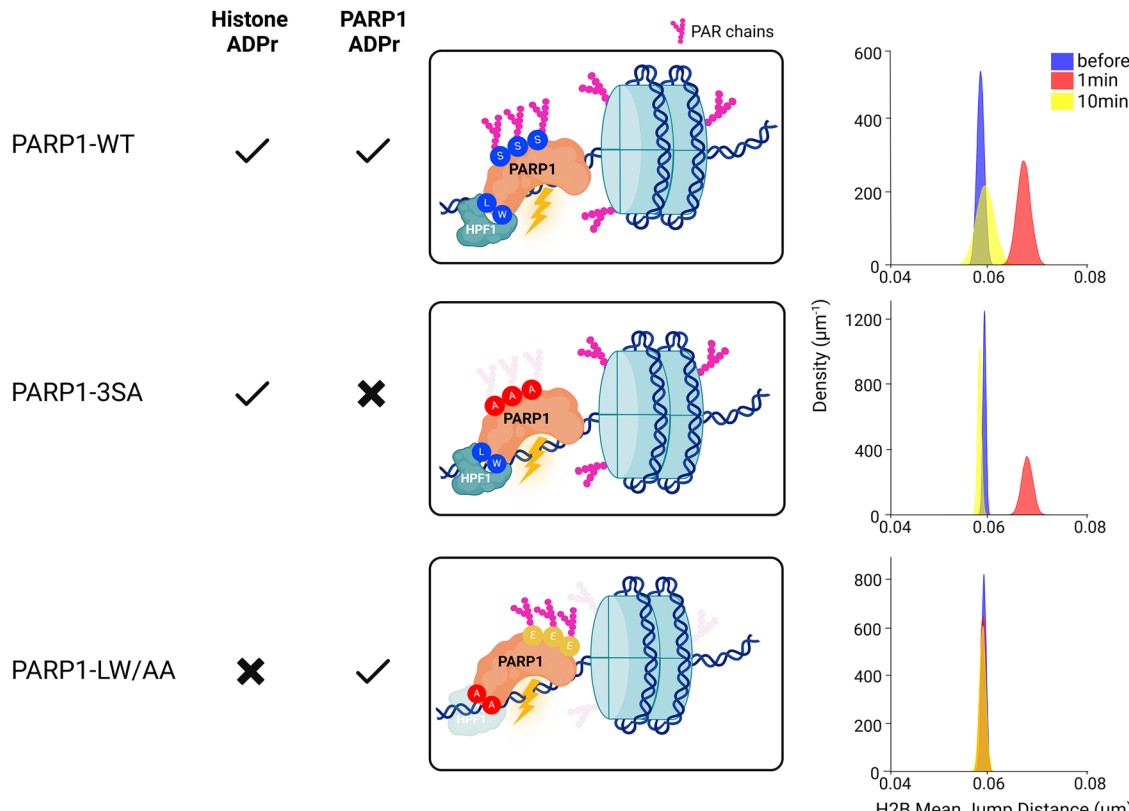

**Fig. 4 | The transient increase in nucleosome dynamics at sites of damage is controlled by histone ADP-ribosylation.** On the left of each panel, a sketch shows the characteristics of PARP1 automodification and histone ADP-ribosylation depending on the PARP1 construct expressed in U2OS PARP1 KO cells. On the right is shown the mean jump distance KDE plots for the immobile population of H2B tracks inside the irradiated region, before and at different times after micro-irradiation at 355 nm. Total number of cells analyzed in three independent experiments ($N$). PARP1-WT $N_{bef}$ = 13, 852 trajectories; $N_{1min}$ = 6, 280 trajectories; $N_{10min}$ = 4, 147 trajectories; PARP1-3SA $N_{bef}$ = 29, 3828 trajectories; $N_{10min}$ = 13, 2457 trajectories; PARP1-LW/AA $N_{bef}$ = 14, 1598 trajectories; $N_{1min}$ = 12, 1043 trajectories; $N_{10min}$ = 8, 972 trajectories. Created in BioRender. Garcia Fernandez, F. (2025) https://BioRender.com/5akfy1x.

recondensation phase which led to a compaction state that is higher than the pre-damage one (Fig. 1). While chromatin opening was shown to be important for facilitating access to DNA lesions (Supplementary Fig. 1B and ref. 21), the recondensation was also proposed to trigger the recruitment of some members of the repair machinery, potentially in relation to transcription shut-down at sites of DNA lesions[40,63,64]. Given the key role played by ADPr signaling during the chromatin decondensation step, we wondered whether this pathway could also regulate the recondensation process.

First, we monitored the kinetics displayed by the ADPr signal to compare them to those of chromatin relaxation (Fig. 1). In agreement with our recent findings[10], we observed that ADPr signal at DNA lesions could be decomposed in an early acute PAR peak and a more progressive and sustained MAR wave (Fig. 5A). The timeframe of these two components suggests that, while the transient PAR surge may trigger chromatin loosening, the maintenance of the open state might be rather controlled by the more persistent MAR signal. To test this hypothesis, we analyzed the impacts of the loss of ARH3. Indeed, this hydrolase, while not regulating the PAR signal, controls the progressive removal of the MAR marks at DNA lesions (Fig. 5A). Importantly, the loss of ARH3 also triggered a possible imbalance in the double-strand break repair pathways as shown by the increased accumulation of the NHEJ-related protein 53BP1 in ARH3 KO cells while the HR-related protein BRCA1 accumulation remained unchanged (Supplementary Fig. 5). Together with increased sensitivity to genotoxic stress observed in ARH3 KO cells[65], these data suggest that the timely removal of MAR signaling by ARH3 contributes to efficient DNA

repair. Regarding chromatin remodeling at sites of damage, we found that, while not affecting decondensation, the loss of ARH3 strongly impaired the recondensation process (Fig. 5B). ARH3 KO cells did not reach the over-condensed state observed in WT cells and were even unable to recover to the pre-damage chromatin compaction level. At the single nucleosome scale, ARH3 KO cells displayed persistent increased nucleosome mobility up to 10 min post-irradiation in contrast to the recovery observed in WT cells (Fig. 5C). These different findings reveal that the erasure of the MAR signal by ARH3 is crucial for the restoration of the chromatin structure following its early relaxation upon damage induction. This provides a functional explanation for the recently revealed temporal bimodality of PARP1 signaling[10] and deepens our understanding of the key role played by PARP1 in the control of chromatin structure at sites of DNA damage.

## Discussion
### Chromatin "breathing" at DNA lesions: a multiscale choreography of chromatin remodeling events
It is now well established that the DDR includes various chromatin remodeling steps, which are crucial for the efficient and faithful restoration of genomic integrity[66]. In yeast, a compelling model has emerged in relation to double-strand break repair, where an increase in chromatin mobility triggered by H2A phosphorylation as well as the homologous recombination machinery, facilitates homology search[25,29,33,34]. The picture remains less clear in mammals with different results depending on the time after damage induction as well as the chromatin landscape in which the lesions occur[3,26,27,67–70]. Importantly,

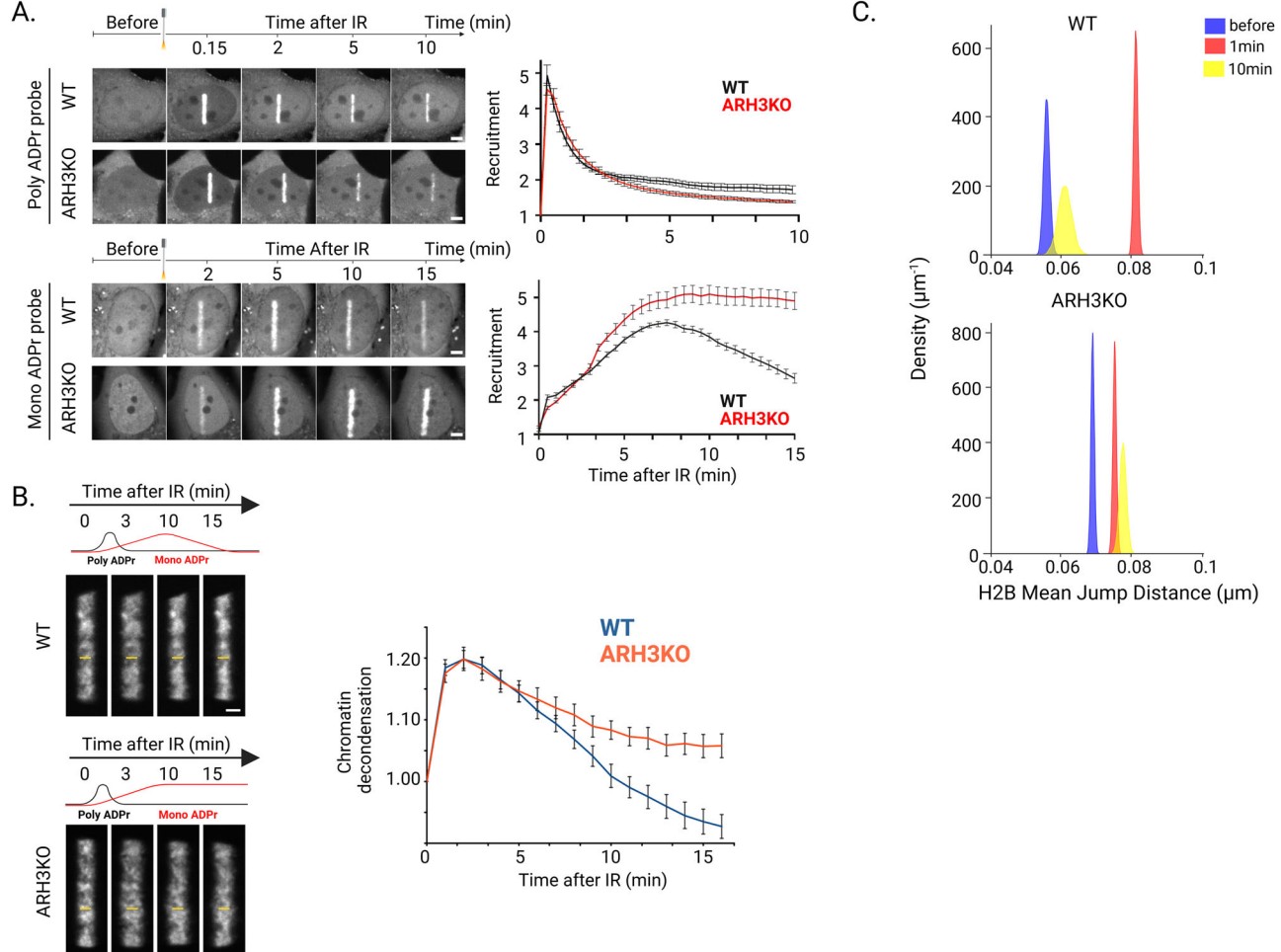

**Fig. 5 | The removal of MAR marks by ARH3 is needed for the recovery of the chromatin state following its initial relaxation at sites of damage.**
**A** Representative confocal images and recruitment kinetics of the GFP-tagged WWE domain of RNF146 (PAR sensor) and Macrodomain of Macro D2 (MAR sensor) expressed WT and ARH3 KO U2OS cells after irradiation at 405 nm. Scale bars: 4 μm (PAR sensor $N_{WT}$ = 12, $N_{KO}$ = 12; MAR sensor $N_{WT}$ = 11, $N_{KO}$ = 12, mean ± SEM.)
**B** Representative confocal images and relative average thickness of the photo-activated damaged area in WT and ARH3 KO cells expressing H2B-PAGFP and irradiated at 405 nm. Scale bars: 2 μm. Curves of the average thickness of the photo-activated line are mean ± SEM (WT $N$ = 12, ARH3 KO $N$ = 16). **C** Mean jump distance KDE plots for the immobile population of H2B tracks inside the irradiated region, before and at different times after micro-irradiation at 355 nm in WT and ARH3 KO U2OS cells. Total number of cells analyzed in three independent experiments ($N$). WT $N_{bef}$ = 23, 847 trajectories total, $N_{1min}$ = 13, 1643 trajectories, $N_{10min}$ = 7, 603 trajectories; ARH3 KO $N_{pre}$ = 21, 1624 trajectories; $N_{1min}$ = 21, 1409 trajectories; $N_{10min}$ = 7, 357 trajectories. Created in BioRender. Garcia Fernandez, F. (2025) https://BioRender.com/5akfy1x.

most of these studies focused on a single spatial scale, which precludes the establishment of a global model for a chromatin structure that is inherently multiscale, sometimes even referred as fractal-like[71]. In this work, we aimed to overcome this technical limitation and enable comprehensive analyses of chromatin behavior by developing an original multiscale framework to assess early changes in the chromatin structure upon DNA damage at multiple levels: from the chromosome scale to the chromatin fiber and down to individual nucleosomes. For the analysis of nucleosome motions, we established a CNN-based approach to classify the single trajectories and extract step distributions (HTC, Histones Trajectory Classifier). While this approach has been optimized for the detection of nucleosomes stably incorporated into chromatin, applying it to study the dynamics of other factors would require a thorough retraining and validation of the classifier, especially for short trajectories. Thanks to the multiscale analysis tool that we developed, we uncovered a "breathing" mechanism that affects the different folding scales of the chromatin immediately after damage induction (Fig. 6), thus modulating DNA accessibility in the vicinity of the breaks. Our findings demonstrate the tight connection between chromatin mobility at the single nucleosome scale and its global compaction state, an aspect for which a unified general model was previously lacking, even beyond the DDR[46,72,73]. By monitoring the precise timing of this remodeling process, our work also reveals that the relationship between the different chromatin folding scales is more than a simple direct correlation. Indeed, while the rapid increase in the mobility of the nucleosomes along the fiber upon DNA damage is associated with a global reorganization, the maintenance of the resulting open chromatin state does not seem to require enhanced nucleosome dynamics. Therefore, the acute surge in nucleosome mobility appears as a transient "activated state" allowing for the switching between two persistent chromatin conformations displaying different compaction levels. Our work illuminates a sophisticated, multifaceted relationship between chromatin folding scales, paving the way for in-depth characterization of the mechanisms underlying the transitions between chromatin states.

**Histone ADPr is both sufficient and necessary to promote multiscale chromatin reorganization at DNA lesions**
In line with several previous reports[19,20,74,75], our work highlights the crucial role played by PARP1 in the regulation of chromatin

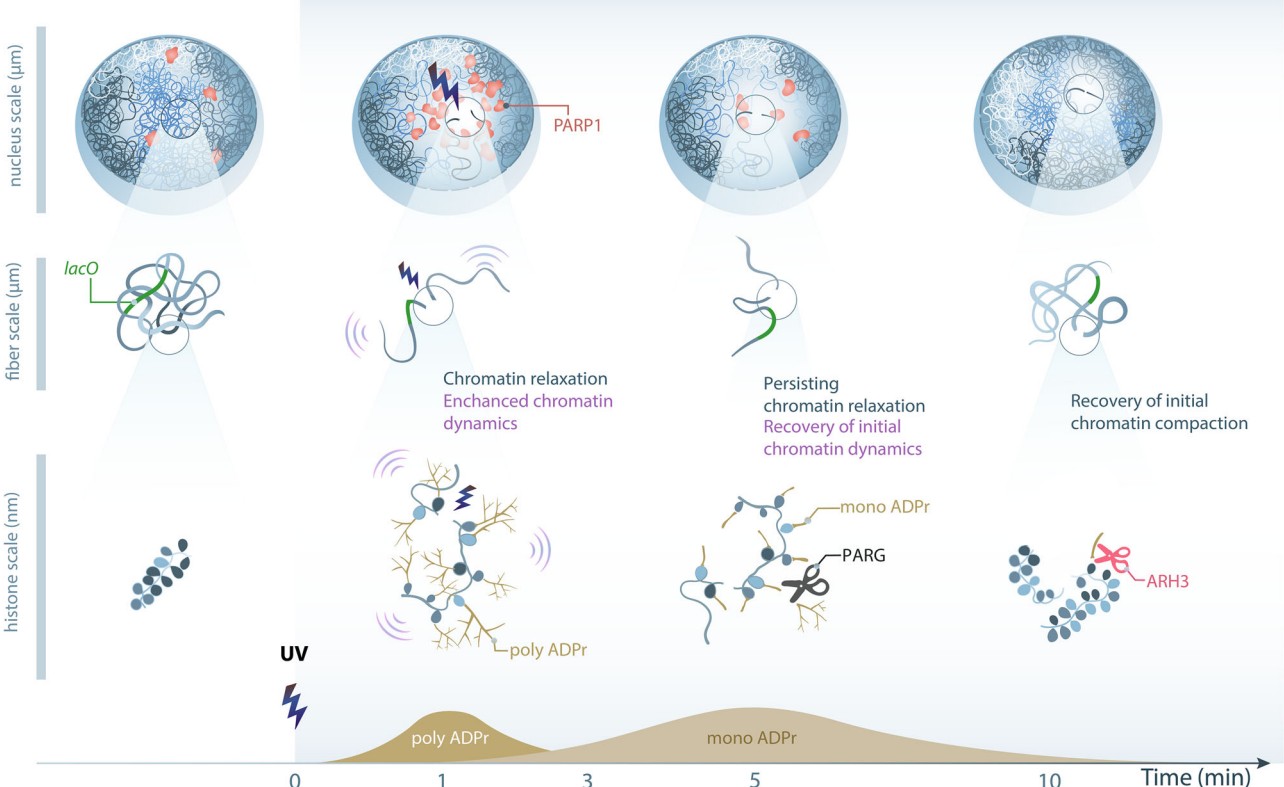

**Fig. 6 | Model of ADPr-dependent multiscale chromatin breathing at sites of DNA damage.** A few seconds after DNA damage, chromatin undergoes rapid decondensation along with an increase of its dynamics at the fiber to the nucleosome scales, a process triggered by histone ADP-ribosylation. While nucleosome dynamics rapidly drops, chromatin remains in an open state for several minutes until MAR erasing by ARH3 allows gradual recondensation (Art by Olga Markova).

conformation. We found that inhibiting PARP1 activity reduced chromatin mobility at DNA lesions, while it seemed to slightly increase it in the absence of exogenous damages. PARP1 is able to dynamically associate with undamaged chromatin, such as with linker DNA, which could modulate chromatin conformation[19]. Yet, the structural features of this interaction[76,77] as well as the subsequent ADPr activity[78], appear to be different from those at DNA lesions. Therefore, PARP1 inhibition may impact the conformation of undamaged chromatin differentially from the damaged one. At DNA lesions, our current and previous findings[21] demonstrate that decorating the chromatin fiber with ADP-ribose marks is itself sufficient to promote its decondensation. This generic process, in coordination with the probably more specific activity of the multiple chromatin remodelers recruited to DNA lesions in an ADP-ribose dependent manner[4,64,79,80], is crucial for the establishment of a repair-competent chromatin conformation in the vicinity of the DNA breaks. Our data, in line with in vitro results on isolated chromatin fibers, indicate that histone ADPr is unlikely to promote a major disruption of the nucleosome architecture leading to eviction and subsequent chromatin unfolding[4,20,75]. Rather, ADPr of linker histone was shown to inhibit its ability to promote chromatin compaction[74]. Therefore, ADPr may change the conformation of the nucleosome at the entry-exit site and lead to a partial eviction of linker histone[22], thus promoting chromatin loosening. Besides affecting nucleosome conformation, ADPr could also act at higher chromatin folding scales by inhibiting nucleosome self-association[75,81]. Negatively charged ADP-ribose chains on the nucleosomes may stiffen the chromatin fiber due to self-repulsion along the polymer, thus promoting reduced packing. This model is in line with the increased nucleosome mobility that we observed at sites of damage which, assuming that these motions can be described by a simple Rouse model, imply an increase in the rigidity of the chromatin fiber[29,49,82]. Given that inter-fiber nucleosome interactions were proposed to dominate over intra-

fiber ones[83] in the nucleus, histone ADPr may also impair fiber-fiber packing, leading to further decrease of the chromatin compaction state.

## MAR marks maintain chromatin in an open conformation at sites of DNA damage

The ADP-ribose signal at sites of DNA damage has historically been considered to be mainly composed of PAR polymers. However, recent technological advances have revealed prevalent MAR marks that exhibit kinetics different from those of PAR, implying distinct roles in PARP1 signaling[10]. Our data indicate that, while the acute PAR wave might be necessary for the initial unfolding of chromatin, the more persistent MAR signal could be sufficient to maintain an open conformation in the vicinity of the DNA lesions (Fig. 6). Therefore, our findings establish distinct functional role for this abundant and enduring MAR signal generated by PARP1. The fact that the relaxation step is not associated with nucleosome disassembly implies that this reorganization can be easily reversed by the removal of the MAR signal along the chromatin fiber by MAR hydrolase ARH3. At low folding scales, MAR also seems to regulate nucleosome mobility as the loss of ARH3 impairs the dampening of nucleosome mobility that rapidly follows its initial surge upon DNA damage induction. This dampening, as well as the subsequent chromatin recondensation, may require the erasure of MAR marks to allow the addition of other modifications on the histone tails restraining nucleosome motions and promoting a closed chromatin conformation. In favor of this hypothesis is the observation that ADPr competes with several other marks on histone tails[81,84,85], with this competition potentially regulating certain aspects of the DDR[86]. We also previously showed that the histone deacetylase HDAC1 gets recruited to DNA lesions via its interaction with the chromatin remodeler CHD7 and that its deacetylase activity regulates the chromatin recondensation step[64]. Therefore, it is possible that

HDAC1 also triggers the dampening of nucleosome motions by erasing acetylation marks on histones. It would be interesting to investigate whether HDAC1 recruitment or activity at DNA lesions is affected by persistent MAR signaling consecutive to the loss of ARH3. This active recondensation process controlled by ARH3 ultimately leads to a chromatin compaction state that appears denser than before damage, which could promote the local inhibition of transcription, a process that contributes to repair efficiency by avoiding collisions between the transcription and repair machineries[87]. Noteworthy, the open chromatin conformation induced by persistent MAR signal on histones may promote pathological unbalances in the transcriptional profiles of patient cells with ARH3 mutations associated with neurodegenerative diseases[88].

## ADPr-dependent chromatin reorganization as a generic regulator of DNA accessibility

DNA wrapped around the nucleosomes shows higher susceptibility to MNase digestion upon histone ADPr[75], suggesting that it became more accessible. At higher folding scales, nucleosome dynamics was proposed as a central regulator of chromatin accessibility in living cells[43,46,89]. Finally, our current and previous work show that the local relaxation of the chromatin controlled by ADPr at DNA lesions increases the binding rates of DNA-binding sensors[5]. Together, these findings draw a compelling picture in which the multiscale impact of histone ADPr on chromatin architecture triggers increased DNA accessibility in the vicinity of the lesions. While this might be a generic way to promote the accumulation of repair factors from different pathways at early stage of the DDR[21], the subsequent recondensation regulated by ARH3 could potentially contribute to repair pathway choice due to the specific retention of a subset of these repair factors (Supplementary Fig. 5 and refs. 40,64). Besides the DDR context, our findings also demonstrate that a dynamic and accessible conformation may be a generic feature of ADP-ribosylated chromatin, independently of the presence of DNA lesions. Given that PARP1 also regulates transcription[90,91], chromatin reorganization triggered by histone ADPr could modulate access to transcription factors[91]. Therefore, our results identify the ADPr signaling as a key regulator of the dynamic chromatin conformation within the nucleus, potentially influencing multiple cellular functions involving DNA transactions.

## Methods

### Plasmids

PmEGFP-PARP1, WT as well as the point mutants S499A/S507A/S519A (3SA) and LW/AA L1013A/W1014A (LW/AA), were previously described[21], as well as pmEGFP-WWE and pmEGFP-Macrodomain of macroD2[21]. pcDNA5/FRT/TO-FLAG-EGFP-BRCA1, pLacI-EGFP, p53BP1-EGFP and H2B-PAGFP were gifts from J. Morris[92], G. Timinszky[93], and J. Ellenberg[94], respectively. To generate the pH2B-HaloTag, we amplified the HaloTag sequence from the plasmid pAT496 (pBS-SK-Halo-KanMX), kindly provided by C. Wu, using primers BshTI_ATG-Halo-Fwd (attaCACCGGTCGCCACCatggcagaaatcggtactgg) and NotI-Stop-End-Halo-Rev (attgcggccGCTTTAggaaatctctagcgtcgacagc) and replaced PAtagRFP[4] in pH2B-PAtagRFP4 using BshTI / NotI.

### Cell culture

All cells used in this study were cultured in DMEM (Sigma) supplemented with 10% FBS, 100 µgml⁻¹ penicillin, and 100 Uml⁻¹ streptomycin and maintained at 37 °C in a 5% CO2 incubator. U2OS WT were obtained from ATCC. U2OS KO for PARP1 and ARH3 cells were kindly provided by I. Ahel[95]. The U2OS 2-6-3 cell line[41] harbors a repetitive array of the *lacO* binding sequence at the chromosomal location 1p36 and was kindly provided by A. Coulon. For transient expression, the GFP-tagged plasmids were transfected with X-tremeGENE HP (Sigma) according to manufacturer instructions. To establish cell lines stably expressing H2B-HaloTag, cells were transfected with the H2B-HaloTag

plasmid and selected using media supplemented with 500 µg.ml⁻¹ G418. The PARP1 inhibitors Talazoparib (Euromedex) were used at 30 µM and added to the cell medium 10 min prior imaging. For PARG inhibition, cells were treated with 25 µM of PDD00017273 (Bio-Techne, USA) for the indicated durations. For HaloTag labeling, cells were incubated for 30 min with 10 nM of HaloTag ligands conjugated to the photoactivatable dye JF549, kindly provided by L. Lavis. For Hoechst presensitization, cells were bathed with culture medium containing 0.3 µg/ml Hoechst 33342 (Sigma) for 1 h. Immediately before imaging, the growth medium was replaced with a $CO_2$-independent imaging medium (phenol red-free Leibovitz's L-15 medium, ThermoFisher). All live-cell experiments were performed on unsynchronized cells.

### Western blotting

Cells were lysed on Triton-X buffer (1% Triton X-100, 100 mM NaCl, 50 mM Tris-HCl, pH 8.0, 5 mM $MgCl_2$, 0.1% Benzonase (Sigma-Aldrich), 1× protease inhibitor (Roche)) on an orbital rotator for 30 min at 4 °C. Samples were centrifuged at $20,000 \times g$ for 15 min, and supernatant was collected. Protein samples were quantified using Bradford (Bio-Rad), and equal amounts of protein were loaded on gels for SDS−PAGE prior to immunoblotting. The membranes were blocked in PBS buffer with 0.1% Tween20 and 5% non-fat dried milk for 1 h at room temperature and incubated overnight at 4 °C with the following primary antibodies: anti-pan-ADPr (MABE1016, Sigma, 1:1500), which binds both MAR and PAR marks, anti-PARP1 (homemade, 1:10000), anti-ARH3 (hpa027104, Sigma, 1:1500), anti-H3 (Ab1731, Abcam, 1:2500). Then, the membranes were incubated with a peroxidase-conjugated secondary anti-rabbit antibody (P039901-2, Agilent, 1:3000) for 1 h. Blots were developed using ECL (Thermo) and analyzed by exposing to films.

### Confocal imaging and quantification

Changes in the chromatin compaction state and protein recruitment at sites of laser irradiation was performed as previously described[4,21]. In brief, images were acquired either on a Ti-E inverted microscope from Nikon equipped with a CSU-X1 spinning-disk head from Yokogawa, a Plan APO 60×/1.4 N.A. Oil-immersion objective lens and a sCMOS ORCA Flash 4.0 camera for Hamamatsu; or on an Olympus Spin SR spinning disc system equipped with a CSU-W1 spinning-disk head from Yokogawa (50 micron pinhole size), a UPLSAPO 100XS/ 1.35 N.A. silicon-immersion objective lens and a sCMOS ORCA Flash 4.0 camera. Laser irradiation of Hoechst-presensitized cells was performed along a 10 or 16 µm-line through the nucleus with a continuous 405 nm laser set at 125–130 mW at the sample level. Chromatin remodeling upon 355 nm irradiation and recruitment of GFP-tagged BRCA1 was monitored on a Zeiss LSM 880 confocal setup equipped with a C-Apo ×40/1.2 N.A. water-immersion objective and a GaAsP detector array for fluorescence detection. Nuclei of non-sensitized cells were irradiated within a region of interest of 100-pixel width and 10-pixel height with a pulsed 355 nm laser (UGA-42 Caliburn, Rapp Optoelectronic) for the chromatin remodeling measurements, or with a Ti:sapphire femtosecond infrared laser (Mai Tai HP, Spectra-Physics) with emission wavelength set to 800 nm, for the BRCA recruitment experiment. For all these live-cell imaging experiments, cells were maintained at 37 °C with a heating chamber. The changes in the chromatin compaction state were measured using a custom MATLAB routine that estimates the thickness of the photo-converted or bleached H2B line relative to its value immediately after damage induction. After segmentation of the damaged chromatin line by k-means thresholding, the line thickness was estimated as the minor axis of an ellipsoid fitting the segmented area. To quantify protein recruitment, the mean fluorescence intensities were evaluated over time within the irradiated area and the whole nucleus, both segmented manually on ImageJ/FIJI or Olympus CellSense. After background subtraction, the intensity in the irradiated area was

divided to the nuclear intensity to correct for imaging photobleaching, and then normalized to the signal prior to DNA damage.

## Single-particle tracking

Dynamics of the *lacO* array and the single H2B proteins were monitored on an inverted Nikon Ti microscope, equipped with an EM-CCD camera (Ixon Ultra 897 Andor) and a 100×/1.4NA or 1.3 NA oil-immersion objective, leading to a pixel size of 160 nm. Cell were maintained at 37 °C using a Tokai device (STXG-TIZWX-SET). A pulsed diode 355 nm laser monomode remotely controlled with the Pangolin Software (LASER ERROL) was coupled to the microscope to allow laser irradiation within a predefined line within the cell nucleus (3.2 μm × 0.4 μm) for 110 ms. The dynamics of the *lacO* array was monitored at a frame rate of 33 Hz. Fluorescent beads (FluoSpheres, ThermoFisher) were used as fiducial markers to correct for cell drift. For the tracking of single H2B-Halo in living cells, the PA-JF549 ligand was photo-activated by the 405 nm laser (1 pulsation every 10 frames, power of 0.006 kW/cm$^2$ at the sample) and excited by the 561 nm laser (continuous excitation, power of 7 kW/cm$^2$ at the sample). Single-molecule imaging sequences of 5000 frames were acquired at a frame rate of 100 Hz. For single molecule detection, position refinement and track reconstruction, we used the SlimFast multi-target tracking algorithm[96]. The parameters used for the tracking are shown in Supplementary Table 1. Home-made routines written in Matlab (Mathworks) were used to visualize the detection density maps and trajectories. H2B dynamics was measured within a rectangle of 3 μm large and whose height was limited by the nucleus border, which was either encompassing the irradiated area or localized away from it. Approximately 1000 trajectories per imaging sequence were monitored within such region of interest, allowing for the building of the mean jump distance kernel density estimation (KDE) plots. The mean single-molecule track length was approximately 170 ms (17 frames), much shorter than the characteristic fluorescence decay of 12.5 s estimated for the PA-JF549 ligand. Therefore, track lengths are limited by out-of-focus movement rather than PA-JF549 photobleaching.

## Mean squared displacement analysis

To characterize the dynamics of the *lacO* array, the time-average mean squared displacement curves were derived from each trajectory as follows:

$$MSD(n \cdot \Delta t) = \frac{1}{N-n} \sum_{i=1}^{N-n} \left[ (x_{i+n} - x_i)^2 + (y_{i+n} - y_i)^2 \right] \quad (1)$$

where N is the total number of points within the trajectory, (x, y) the coordinates of the locus in 2-dimensions and Δt the time interval used during the acquisition. To obtain a precise estimation of the fitted parameters, we calculated time-ensemble-averaged MSD over several trajectories, which are simply referred to as "MSD" in the Figures using a home-made Matlab code[49]. In line with previous studies[49,82,97], the mean MSD curves were fitted with the following anomalous diffusion model:

$$MSD(t) = At^\alpha + \sigma^2 \quad (2)$$

where $\alpha$ is the anomalous exponent, A the anomalous diffusion coefficient and $\sigma$ the positioning accuracy. Here, we found a better agreement with the anomalous diffusion model, consistent with previous studies[49,82,97]. $\alpha < 1$ corresponds to subdiffusive dynamics, referring to tracked objects that reiteratively scans neighboring regions before reaching a distant position[98]. In contrast, $\alpha > 1$ corresponds to motions displaying a directed component[99]. The anomalous diffusion coefficient A quantifies motion amplitude. It is proportional to the diffusion coefficient only in the case of normal diffusion ($\alpha = 1$), which is rarely observed in biological systems. Besides this analysis of

diffusion anomaly, we also quantified locus mobility with an effective diffusion coefficient D$_{lacO}$ calculated as D = p/4 where $p$ is the slope of the linear fit of the first 4 points of the MSD curves. To compare with the diffusion of the slow population of H2B, we also extracted an effective diffusion coefficient D$_{H2B}$. Since H2B trajectories are much shorter than *lacO*, to reduce the experimental and localization noise, we calculated a denoised averaged diffusion coefficient by measuring the tangents of the fitted MSD curve between 0.1 s and 1.0 s.

## Classification of the H2B tracks

We used a CNN[100,101] to classify H2B trajectories into 3 categories: immobile, hybrid, and mobile (Supplementary Fig. S2E–G). In the preprocessing step, the single-molecule trajectories are converted into 2D images of 512 × 512 pixels to consider the sub-pixel accuracy of localization. Displacements between two consecutive positions are interpolated as a straight segment. To improve classification accuracy, each segment is given a third dimension, defined as a color. The segment color is chosen according to the instantaneous diffusion coefficient calculated between the corresponding two positions, gathering them into three heuristically chosen ranges: 0 μm$^2$/s <red ≤ 0.5 μm$^2$/s, 0.5 μm$^2$/s <green ≤ 1 μm$^2$/s, 1 μm$^2$/s <blue, respectively. For the training of CNN model, starting from 3120 manually annotated images of trajectories (1040 for each class), the annotated trajectories are standardized with a subtraction of the first coordinate for each individual trajectory. Then, trajectories are augmented with rotation, 23 times of rotation with increasing 24/Pi radian for each individual trajectory, to generate a set of 71,760 augmented images. Note that this rotation is done at coordinate-level instead of image-level to avoid image distortion. After random shuffling, 80% of the augmented images were composing our training set while the remaining 20% were used as a validation set. The structure of CNN model is as follows: 512 × 512 input size with 5 convolutional layers (C1: 8 × 8–32, C2: 5 × 5–64, C3: 2 × 2–128, C4: 2 × 2–256, C5: 2 × 2–512) where the corresponding numbers are the kernel size and the number of filters for each convolutional layer. Each convolutional layer is followed by max pooling, batch normalization layers with ReLU activation. The total number of trainable parameters is 762 K. The optimization is performed with[102], including cross-entropy loss. The softmax function is used for 3-classification at the end of the network. The learning rate and batch size for the training are 0.0005 and 16, respectively. The trained model shows around 98% accuracy on the validation set of our annotated data (Supplementary Fig. 2E, F). An additional validation of CNN on simulated data displaying fractional Brownian motions mimicking histone dynamics as also been performed. The output of this validation as well as exemplary training images are available publicly on [https://github.com/JunwooParkSaribu/HTC/blob/main/validation/Validation_process.ipynb]. We also compared the results of our classifier with the outputs of the vbSPT algorithm[56]. vbSPT classifies individual segments within trajectories rather than the full traces. Therefore, to compare this algorithm with our CNN-based approach, we considered that a given trajectory belonged to the immobile or mobile class if all the segments therein were classified as immobile or mobile, respectively. Trajectories with segments switching between classes were classified as hybrid.

## Representation of histones displacements

After calculation of the mean jump distance of each trace belonging to the different classes, we applied a bootstrapping step to build the averages of mean jump distance distributions. The mean jump-distance of molecules should follow a Gaussian distribution according to the central limit theorem, assuming it is independent and identically distributed. However, errors during track reconstruction and trajectory mis-classification may distort the empirical distribution, potentially driving inappropriate biological conclusions.

Therefore, we used bootstrapping to compute the confidence intervals of averaged mean jump-distances and to approximate the underlying distribution for each condition. We resampled 10,000 averaged mean jump-distances for each condition pre- and post-damage and estimated confidence intervals for each of the boot-strapped distributions. The averages of mean jump-distance distributions obtained from the bootstrapping are shown as KDE plots. We used the SciPy package of Python[103] to obtain the bootstrap distributions and the probability density function plots with kernel density estimation using Scott's rule[104]. All the averaged mean jump distances were equally weighted in KDE. All the parameters used for the classifiers and the estimation of mean jump distributions are shown in Supplementary Table 1.

### Reporting summary
Further information on research design is available in the Nature Portfolio Reporting Summary linked to this article.

## Data availability
The data that support the findings of this study are openly available in figshare at https://doi.org/10.6084/m9.figshare.28458161. Source data are provided with this paper.

## Code availability
The code for the Histone Trajectory Classifier HTC is available here: https://doi.org/10.5281/zenodo.13334706 Contact: junwoo.park@sorbonne-universite.fr for further information.

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

## Acknowledgements

We thank C. Maison, D. Bailly, A. Forest (Institut Curie), and C. Fouquet (Institut de Biologie Paris Seine, Sorbonne University) for their help with the cell culture. The authors also thank the PICT-IBiSA@Pasteur Imaging Facility of the Institut Curie, a member of the France Bioimaging National Infrastructure (ANR-24-INBS-0005 FBI BIOGEN), C. Chaumeton, and F. Lam from the cellular Imaging Facility (Institut de Biologie Paris Seine, Sorbonne Université). We thank the Microscopy-Rennes Imaging Center (BIOSIT, Université de Rennes), member of the national infrastructure France-BioImaging supported by the French National Research Agency (ANR-24-INBS-0005 FBI BIOGEN), for providing access to the imaging setups, as well as S. Dutertre, X. Pinson, and G. Le Marchand for technical assistance on the microscopes. We thank J. Morris, G. Timinszky, J. Ellenberg, S. Buratowski, A. Coulon, and L. Lavis for sharing reagents. We are grateful to E. Fabre for her fruitful comments on the manuscript and A. Mansuy for sharing his software expertise. Illustrations in figures were created in BioRender. Garcia Fernandez, F. (2025) https://BioRender.com/5akfy1x. The S.H.'s and J.M.H.'s groups received financial support from the Agence Nationale de la Recherche (ANR-18-CE12-0015-03 RepairChrom and ANR-22-CE12-0039 AROSE). The J.M-H team was financially supported by the i-Bio Initiative from the Idex Sorbonne University Alliance, the IBPS Incentive Action and the ATIP Avenir 2021. I.M.'s lab was funded by the Max Planck Society, the Deutsche Forschungsgemeinschaft (DFG, German Research Foundation) under Germany´s Excellence Strategy (CECAD, EXC 2030-390661388) and by the European Research Council (ERC-CoG-864117). I.M. and E.J.L. received support from the EMBO Young Investigator Program and the Cologne Graduate School of Ageing Research, respectively. This work has inspired part of the Muse-IC project, a collaborative project between musicians and composers aiming to create musical pieces inspired by recent scientific discoveries.

## Author contributions

Data curation: F.G.F., E.P.J., V.I., R.S., E.J.L., S.H., J.M.H. Formal analysis: F.G.F., J.P., S.H., J.M.H. Methodology: F.G.F., J.P., C.H., S.H., J.M.H. Writing, original draft: F.G.F., J.P., S.H., J.M.H. Writing, review and editing: F.G.F., J.P., S.H., J.M.H. Software: J.P. Cell line generation: C.C. Funding: A.T., N.S., I.M., S.H., J.M.H. Conceptualization: C.H., S.H., J.L.H. Supervision: N.S., I.M., S.H., J.M.H. Project design: S.H., J.M.H.

## Competing interests

The authors declare no competing interests
