## [Peer Review file · Nature Communications]

Single-nucleosome imaging reveals principles of transient multiscale chromatin reorganization triggered by histone ADP-ribosylation at DNA lesions.

Corresponding Author: Dr Judith Miné-Hattab

Version 0:

Reviewer comments:

Reviewer #1

(Remarks to the Author)

How the DNA damage response is involved in the remodeling processes of chromatin remains unclear. By combining micro-irradiation with live-cell multiscale imaging, García Fernández et al. found a temporary increase in nucleosome mobility and alteration of chromatin from a densely packed state to a looser conformation within the first seconds after DNA damage, making it accessible to the repair machinery. The authors demonstrated that histone poly-ADP-ribosylation is required to trigger this switch, and mono-ADP-ribosylation maintains the open-chromatin state. Once these histone marks were removed by the ARH3 hydrolase, chromatin recondensed. Their convincing data and intriguing findings address how PARP and ADP-ribosylation contribute to DNA damage repair. I would support this paper. For the publication in Nature Communications, my specific comments are as follows:

Major comments:

(1) When the H2B-HaloTag trajectories were categorized using CNN, the authors used 80% of trajectories for training. They should clarify how they categorized the trajectories into the three during training. And how did they determine the boundaries, for instance, " $0 \mu\text{m}^2/\text{s} < \text{red} \leq 0.5 \mu\text{m}^2/\text{s}$ "?

It is also helpful to use other categorization methods, such as vbSPT (PMID: 23396281).

(2) Fig. 1C, right "inside damage" panel. Why did the blue one (before IR) have the three peaks? After 12 min, the gray one shifted more to the left than the blue one. Is this significantly different? Any reason?

(3) I wonder whether there are any differences in nucleosome response between euchromatin and heterochromatin. Could the author look at nucleosome response around the nuclear periphery upon damage? This can be an excellent addition to the paper.

(4) After chromatin recondensation (~10 min), does accessibility to damage sensor proteins such as BZIP decrease?

Minor comments

(1) "IR near LacO" in Fig. 1B. The authors could not focus on LacO. Is this because IR photobleach the LacI-GFP signals?

(2) Fig. 1A, "H2B-GFP" should be "H2B-PAGFP".

(3) Page 4, line 7, "While, at short time scales ($t < 5 \text{ min}$)..." Should " $t < 5 \text{ min}$ " be changed to " $t < 5 \text{ s}$ "?

(4) Fig. 3C. Chromatin motion was further increased in ARH3 KO +PARGi cells under non-irradiation. Why? And what happens in ARH3 KO +PARGi cells under irradiation?

(5) Page 10, Line 3. "At higher folding scales, nucleosome dynamics was proposed as..." A new modeling data is in a new preprint:

<https://doi.org/10.1101/2024.10.20.618801>

(Remarks on code availability)

Reviewer #2

(Remarks to the Author)

In this manuscript, Fernandez and colleagues use single-nucleosome imaging combined with microirradiation to reveal that ADP-ribosylation (ADPr) of histones at DNA lesions induces transient chromatin decondensation, facilitating DNA repair. By observing chromatin at multiple scales, the authors show that PAR and MAR ADPr signals play distinct roles in initiating and maintaining chromatin decondensation. PAR deposition decondenses chromatin and increases nucleosome mobility, while MAR removal by ARH3 leads to chromatin recondensation. The findings highlight ADPr as a critical regulator of chromatin dynamics during the DNA damage response, enhancing our understanding of chromatin motion at DNA lesions.

While diffraction-limited imaging has been commonly applied to measure chromatin decondensation in response to DNA damage, to my knowledge, the application of single nucleosome tracking together with this assay is new. Although I am not an expert in the DNA damage response and thus cannot comment on the novelty or significance of the findings regarding the temporal destabilization/restabilization of nucleosome motion relative to chromatin condensation, to me the results were interesting. However, I feel that the manuscript could benefit from additional technical validation, particularly with respect to the neural network classifier and consistency in the histograms of the single-molecule jump distances prior to publication.

Major comments:

- The histograms of H2B jump distances appear very odd to me with multiple peaks that suggest separate sub-populations. However, I suspect that since these peaks seem to vary considerably across experiments and the histograms are presented as a continuous rather than binned distribution, it may just be that the histogram bins are too small to capture the consistent trends in the data. Please specify the bin size and indicate what guidelines were used in choosing the bin width. Please also include some metric of cellular and experimental variability in the datasets. While the authors list the number of cells measured for each condition, it's not clear if these came from independent experiments and all the data appear to be aggregated into the histogram. For clarity, an experiment (or cell-by-cell) breakdown of the histograms could be presented in the supplementary materials.
- The implementation of a convolutional neural network to classify trajectories requires additional validation and documentation. It's unclear what the classifier was specifically trained on—were the linked trajectories used to generate synthetic images, which were then annotated into groups? Could the authors clarify this process as well as the overall network architecture? Additionally, I am uncertain why the authors chose this methodology over more established approaches, such as vbSPT (PMID: 23396281), which are commonly used for classifying trajectories or Spot-On (29300163) to determine the fraction of molecules in different diffusive states. Classifying molecules transitioning between states has conventionally been an intractable task for shorter trajectories due to the probabilistic nature of diffusion. It seems that the authors could have attained most of the biological conclusions without developing this new classifier, but if they wish to include it, it needs additional validation and comparison to the other more commonly used methods referenced above. I'd also be curious whether the other more established methods mentioned above confirm the findings from the new CNN.
- The statement "While PARPi treatment slightly impacts the dynamics of the lacO array and the nucleosome in the absence of damage" seems to conflict with the trends in the data provided. I agree that PARPi has minimal effects on the LacI in absence of irradiation. However, the effects on nucleosome motion in the PARPi condition are considerable. This can be seen in comparing blue distributions in Fig2A vs. Fig2B and those in FigS2C where PARPi treatment seems to cause a fairly dramatic increase in nucleosome mobility even without irradiation. In the GFP-LacI PARPi+/IR+ experiments (Supplementary Figure 2), there is a clear decrease in locus dynamics upon induction of DNA damage. For the nucleosomes PARPi+/IR+ experiments (Figure 2), there is an increase in steady state dynamics (comparing the blue "before" distributions in panels A, B, and C) and a loss of the transient increase in dynamics compared upon induction of DNA damage (comparing the red "1 min" distributions in panels A, B, and C). While I understand that the locus tracking and single-nucleosome imaging approaches do not measure exactly the same thing, it would be helpful for the authors to acknowledge and interpret these observed differences. To me, the data suggests that PARPi leads to a dramatic stabilization of lacI locus dynamics within 1 min after DNA damage while not affecting steady state locus dynamics. For nucleosomes, PARPi seems to destabilize nucleosomes at steady state, but prevent the additional increase in nucleosome dynamics after DNA damage. I'm unclear how these results fit into the model proposed in the paper.

Minor comments:

- The description of the chromatin compaction state measurement methods is unclear. It would be helpful for the authors to provide more detail on how the thickness of the photo-converted H2B line was estimated? Specifically, did they employ manual annotation, fit a Gaussian or another model, or use a different approach? Further elaboration on this step would greatly improve reproducibility.
- In general, the authors employ somewhat imprecise terminology when describing chromatin. For example, there are numerous occasions wherein the authors utilize the term "chromatin folding" when referencing their results. For me, this is confusing. While I agree that the increase in the activated stipe width after DNA damage could suggest a chromatin decondensation, I'd hesitate to say anything about chromatin "folding" which suggests some change in how DNA wraps around nucleosomes or in which nucleosomes assemble into some higher-order structure. For me, this isn't supported by the type of measurement performed in the data.

I'm also confused by the usage of the phrase 'chromatin fiber'. The authors occasionally refer to mesoscale chromatin as 'chromatin fiber,' such as in the passage, '...most likely corresponds to H2B proteins stably incorporated into nucleosomes as they display an effective diffusion coefficient similar to that of the chromatin fiber.' However, in other instances, they use 'chromatin fiber' to refer to individual nucleosomes, as in: 'While the expression of wild-type PARP1 or PARP1-3SA in PARP1 KO cells both rescued the transient increase in nucleosome mobility at sites of DNA lesions, this was not the case for PARP1-LW/AA (Figure 4). These data demonstrate that it is the ADPr of histone and not PARP1 that triggers the increase in chromatin fiber mobility at sites of DNA breaks.'

Additionally, in the discussion the authors state "persistent MAR signal could be sufficient to maintain an open conformation in the vicinity of the DNA lesions". The term, "open conformation" is unclear here. I believe the authors are referring to the role in persistent MAR in preventing the recondensation of chromatin as indicated by shrinking in the width of the photoactivated line after DNA damage. To me, this suggests that MAR maintains a decondensed state rather than an "open conformation" which I would associate more with something like MNase accessibility. There may be other studies that relate histone ADPr to DNA accessibility (e.g. via MNase seq), but the data in this manuscript doesn't measure "open-ness" vs. "closed-ness". I feel like it's important to be precise and consistent in the language used to describe specific results.

- In the results, the authors state that "In contrast, chromatin remains in this decompacted state for several minutes despite the rapid drop in nucleosome dynamics". I'm curious what biological process is responsible for stabilizing the nucleosome motion in the decondensed chromatin state? It might be worth revisiting this finding in the discussion section.
- In the methods section for classification of the H2B tracks, the authors refer to a "convoluted neural network (CNN)" I believe that this should be "convolutional neural network".

(Remarks on code availability)

Reviewer #4

(Remarks to the Author)

The authors present an interesting new take on understanding changes to chromatin at sites of DNA damage. Moreover, they show how this movement is regulated by waves of PARylation and MARYlation. Together, their study establishes a new model how ADP-ribosylation controls nucleosome mobility, triggering a transient breathing of chromatin, crucial for initiating the DNA damage response. This study is timely and will be of great interest to the scientists working in the genome stability and chromatin fields of research.

Specific comments:

In figure 1A the authors use 405 nm irradiation to induce DNA damage, which clearly induces large scale chromatin relaxation. In figure 1B and C however, the authors switch to using 355 nm irradiation. Do the authors see the same relaxation kinetic with 355 nm laser as with 405 nm laser?

Do you see any discernible difference in histone movement in the middle of your irradiated line as compared to the edges of the stripe? Or do you see equal movement of histones across the whole stripe? Are there areas of chromatin compaction at the edges of stripes?

Did you compare chromatin movement in damaged cells treated with PARG inhibitors?

Have the authors had the opportunity to compare chromatin dynamics or potential chromatin disassembly with over expression of chromatin remodellers – for example, what happens with the PAR-dependent chromatin remodeler ALC1?

Introduction. 'While PARG is the most active PAR hydrolase, ARH3 is a specific serine MAR eraser.' It would be worth also specifying here that the serine-linked ADP-ribosylation is the most robust form of ADP-ribosylation in DNA damage response and that the histones are the major targets (doi: 10.7554/eLife.34334).

(Remarks on code availability)

Version 1:

Reviewer comments:

Reviewer #1

(Remarks to the Author)

The authors addressed my comments with additional data. I would support this revised paper for publication.

(Remarks on code availability)

Reviewer #2

(Remarks to the Author)

In this revised manuscript, Fernandez and colleagues have addressed many of my (and the other reviewer's comments). I commend the authors for including additional explanation and discussion regarding the neural network classifier and the display of the single molecule tracking data. Overall, I feel that the manuscript is a useful contribution to the field with interesting and careful experimental results. I believe it is likely suitable for publication in Nature Communications provided that the authors can address my remaining concerns detailed below.

1) I still have concerns about how the single molecule tracking data is presented throughout the manuscript, which is very unusual compared to most single molecule tracking/imaging experiments in most of the literature. Specifically, in the revised manuscript, the authors have clarified how the CNN was trained by using the trajectories to generate artificial "images" which can then be classified using convolutional filters. This is an interesting approach and could be useful for future studies. However, my primary remaining concern is that the network was trained on "manually annotated" examples of real molecule trajectories. I suppose for clear cases like H2B where the slow diffusing state is substantially different from the free diffusing state, this may still be fairly accurate. And the fact that vbSPT provides similar results is assuring. However, it does mean that the CNN is essentially being trained to match a subjective classification that may vary from user to user and likely would not generalize well for settings where molecular motion can be more complex (e.g. 3-state diffusion for transcription factors that display both specific and non-specific binding) that cannot be easily classified by eye. Given that quantifying the number of molecules in different diffusive states is a relatively minor part of the manuscript, I would still be in favor of publication provided that the authors include a clear description of these limitations in the manuscript discussion.

2) My other concern is how the jump-length distributions are presented in the revised manuscript. If I understand correctly based on the updated methods section, for most of the figures in the paper, the authors are plotting the kernel density estimator for a bootstrapped analysis on the mean jump distance after filtering for the stable population of trajectories using their CNN classifier. Is this correct? While it may still be "correct" to justify the authors claims, it is very different from how this data is usually displayed. It should be made clear in the main text that the authors are fitting the expected mean jump distance for just the stable population of H2B trajectories after filtering. The authors also need to provide details in the methods section for the fitting procedure used to generate the KDE (what software/function) from the filtered mean jump distances.

Minor comments:

1) First paragraph of the "Results" section the authors state: "U2OS cells expressing H2B fused to the photo-activatable dyes PAGFP". PA-GFP is a protein, not a dye. Please also spell out the acronym upon first usage in the text.

2) The green MSD curve at 10 min after IR in Figure 1B seems to show an upward "kink" at 10 seconds. Based on the error bars, this is remarkably repeatable between experiments. Can the authors comment on or explain this phenomenon?

3) Supplementary Figure 3: The color coding in Figure panel A is switched between the trajectory map and the plot to the right.

4) Supplementary Figure 4A: Please clarify how PARP1 recruitment was measured. The legend indicates WT U2OS cells, but I'm assuming the cells were either transfected or stably expressing some form of fluorescently labeled PARP1?

5) Figure 2A is missing a callout in the main text.

(Remarks on code availability)

Reviewer #4

(Remarks to the Author)

The authors have fully and thoroughly addressed my comments.

(Remarks on code availability)

Version 2:

Reviewer comments:

Reviewer #2

(Remarks to the Author)

The authors have addressed all of my concerns. The manuscript is ready for publication.

(Remarks on code availability)

POINT BY POINT RESPONSE

Point by point response to the comments of the referees regarding the manuscript by García Fernández et al.

We thank the referees for their thorough evaluation of our work. We performed additional experiments and revised the analysis of the single molecule tracks to address their concerns. Our point-by-point response to the specific concerns of the referees can be found below in blue.

Reviewer #1 (Remarks to the Author):

How the DNA damage response is involved in the remodeling processes of chromatin remains unclear. By combining micro-irradiation with live-cell multiscale imaging, García Fernández et al. found a temporary increase in nucleosome mobility and alteration of chromatin from a densely packed state to a looser conformation within the first seconds after DNA damage, making it accessible to the repair machinery. The authors demonstrated that histone poly-ADP-ribosylation is required to trigger this switch, and mono-ADP-ribosylation maintains the open-chromatin state. Once these histone marks were removed by the ARH3 hydrolase, chromatin recondensed. Their convincing data and intriguing findings address how PARP and ADP-ribosylation contribute to DNA damage repair. I would support this paper. For the publication in Nature Communications, my specific comments are as follows:

Major comments:

(1) When the H2B-HaloTag trajectories were categorized using CNN, the authors used 80% of trajectories for training. They should clarify how they categorized the trajectories into the three during training. And how did they determine the boundaries, for instance, " $0 \mu\text{m}^2/\text{s} < \text{red} \leq 0.5 \mu\text{m}^2/\text{s}$ "?

In the revised version of the manuscript, we clarified our approach to categorize the trajectories (see Methods, p 16-17). The histones classifier HTC developed in our work is available here: <https://github.com/JunwooParkSaribu/HTC>. We included a documentation explaining the program as well as a validation using simulations (https://github.com/JunwooParkSaribu/HTC/blob/main/validation/Validation_process.ipynb). HTC aims at classifying trajectories as a whole. Prior to classification, each trajectory was converted into a trajectory-image. Additionally, to improve the classification, each segment along the trajectory-image was color-coded based on the instantaneous diffusion coefficient: $0 \mu\text{m}^2/\text{s} < \text{red} \leq 0.5 \mu\text{m}^2/\text{s}$, $0.5 \mu\text{m}^2/\text{s} < \text{green} \leq 1 \mu\text{m}^2/\text{s}$, $\text{blue} > 1 \mu\text{m}^2/\text{s}$. These boundaries were chosen heuristically by visual inspection of the histone trajectory maps within the nuclei. While for molecules visually appearing as immobile most segments were showing instantaneous diffusion coefficients below $0.5 \mu\text{m}^2/\text{s}$, the segments of the trajectories corresponding to molecules appearing as mobile were usually above $1 \mu\text{m}^2/\text{s}$. HTC was trained with manually annotated data. A set of 1040 immobile, 1040 mobiles and 1040 hybrid traces were selected. Typical examples of the annotated traces from each class are shown in supplementary figure 2G as well as in the above link. We now provide all the annotated data as supplementary data (see `supp_training_sample.zip`) as well as in the github page of HTC (https://github.com/JunwooParkSaribu/HTC/tree/main/validation/training_data). After data augmentation allowing to reach a set of 71,760 images, 80% of them were randomly chosen for the training while the remaining 20% were used as a validation set.

It is also helpful to use other categorization methods, such as vbSPT (PMID: 23396281).

Following reviewer suggestion, we compared our classification approach to the vbSPT algorithm. This gave us the opportunity to improve our analysis by including a bootstrapping step similar to vbSPT.

According to the central limit theorem, the mean jump-distances of the single molecule should follow a Gaussian distribution assuming they are independent and identically distributed. However, noise introduced by localization errors and mis-linking during trajectory reconstruction *via* the tracking algorithm, as well as trajectory misclassification by HTC, may distort the empirical mean jump-distance distributions, potentially driving inappropriate biological conclusions. Therefore, we used bootstrapping to compute the confidence intervals of averaged mean jump-distances and to approximate the underlying distribution for each condition. We resampled 10,000 averaged mean jump-distances for each condition pre and post-damage induction and estimated confidence intervals for each of the bootstrapped distributions. We obtained 95% confidence intervals of $[0.0532, 0.0556]$, $[0.0773 \mu\text{m}, 0.0795 \mu\text{m}]$, $[0.0614 \mu\text{m}, 0.0654 \mu\text{m}]$, $[0.0553 \mu\text{m}, 0.0605 \mu\text{m}]$ and $[0.0542 \mu\text{m}, 0.0572 \mu\text{m}]$ for the bootstrapped distributions before damage and 1 min, 5 min, 10 min and 12 min post-damage, respectively (see revised Figure 1C). The low variance of the bootstrapped distributions shows the strong sample consistency for each condition, which allows for a precise analysis of the changes affecting histone mobility at different times post irradiation.

Then, we compared the results obtained with our HTC method with the outputs of vbSPT when analyzing histone mobility in the immobile category before and 1 min after damage (revised supplementary figure 2I). A fundamental difference between the two methods

is that HTC does the motion classification at the level of the whole trajectory whereas vbSPT does it at the level of single displacements between two consecutive timepoints. Then, to allocate a whole trajectory to a given class using vbSPT, it is necessary to set additional conditions in case all the segments do not belong to the same class. This is complicated by the fact that some jumps arise from localisation errors as well as trajectory mislinking. Here, we defined a trajectory as immobile or mobile only if all the segments were belonging to the corresponding class according to the vbSPT analysis. Trajectories displaying segments of different classes were categorized as hybrid. Due to this conservative definition, the immobile class obtained from vbSPT displayed distributions of mean jump-distances shifted to lower values compared to the output of the HTC approach. Nevertheless, both methods showed a dramatic increase in the mobility of the immobile class upon DNA damage, demonstrating the robustness of this finding (supplementary figure 2I).

(2) Fig. 1C, right "inside damage" panel. Why did the blue one (before IR) have the three peaks?

The data shown on Figure 1C correspond to the mean jump-distance for the trajectories classified as immobile by the HTC algorithm. These distributions gather trajectories from 4 independent experiments performed at different days, corresponding to a total of 1501 traces from 41 cells. The number of independent experiments used to build the distributions is now mentioned in the legend of the figure. Additionally, we show below on figure R1 the breakdown distribution of the mean-distance prior damage induction for each individual experiment. This shows that the three peaks shifted by a few nanometers in the global distribution originate from day-to-day variability rather than subpopulations within the immobile category. Importantly, with the bootstrap step that we included in the revised version of our analysis in line with published SPT analysis tools such as vbSPT, these three peaks are not noticeable anymore on the global mean jump-distance distribution prior to DNA damage (see revised Figure 1C).

Figure R1: Heterogeneity of the H2B jump-distance distribution. Day-by-day breakdown of the mean jump distance distribution for the immobile population of H2B tracks prior to laser irradiation.

After 12 min, the gray one shifted more to the left than the blue one. Is this significantly different? Any reason?

In the rebuttal Figure R2 below, we compared the jump-distance distribution before and 12 min after irradiation in three independent experiments. These data show a consistent and significant decrease in H2B motion 12 min after irradiation compared to the predamage dynamics. This decrease correlates in time with the over-compaction of the chromatin observed in Figure 1A. The experiments described on Figure 5 indicate that the establishment of this over-compacted state associated with reduced histone mobility requires the efficient clearance of the ADP-ribose marks on the histones. Yet, further experiments would be needed to study whether this clearance step is associated with the addition of new histone marks favoring a closed chromatin conformation.

Figure R2: Day-to-day breakdown of the mean jump distance distributions for the immobile population of H2B tracks before (blue) and 12 minutes after micro-irradiation (gray). For each experiment, we observed a significant difference between the two distributions ($p < 0.001$, calculated from Yuen-Welch Test).

(3) I wonder whether there are any differences in nucleosome response between euchromatin and heterochromatin. Could the author look at nucleosome response around the nuclear periphery upon damage? This can be an excellent addition to the paper.

As requested by this reviewer, we compared histone dynamics in the middle and at the periphery of the nucleus within the irradiated line (see revised supplementary figure 3). Prior to DNA damage, H2B dynamics was decreased at the periphery compared to the nuclear center, in line with previous reports (see e.g. PMID: 30824489, 28712725). One minute post-irradiation, we observed a similar relative increase in histone motions in both nuclear regions. These data indicate that histone dynamics is affected similarly upon DNA damage in the heterochromatic rim and in the more euchromatic nuclear center.

(4) After chromatin recondensation (~ 10 min), does accessibility to damage sensor proteins such as BZIP decrease?

In a previous report, we demonstrated that the recruitment of DNA binding domains such as the BZIP domain from C/EBP α at DNA lesions can be used as a proxy to assess DNA accessibility in damaged chromatin (PMID: 31566235). In the revised version of the current manuscript, we monitored the recruitment of BZIP for 15 min post-irradiation and found that the initial accumulation of the sensor at the lesions, which peaks at ~ 3 min post-irradiation, is followed by a progressive release (see new supplementary figure 1B). The kinetics of these two phases closely matches those of the transient opening of the chromatin structure (Figure 1A), suggesting that, while the initial decondensation is associated with increased DNA accessibility, the progressive recondensation of the chromatin restrains it. Nevertheless, at the late stages when chromatin tends to

overcondense compared to its predamage state, we still observed some slight accumulation of BZIP, suggesting that DNA remains more accessible even in this overcompacted chromatin conformation. Future experiments will be needed to investigate further the structural characteristics of this chromatin state.

Minor comments

(1) "IR near LacO" in Fig. 1B. The authors could not focus on LacO. Is this because IR photobleach the LacI-GFP signals?

As hinted by this referee, laser irradiation right at the LacO array photobleaches the GFP signal, precluding further tracking. Therefore we performed the laser irradiation at about 1 μm away from the array.

Figure R3: Microscopy images of a U2OS nucleus harboring a LacI-GFP fluorescent spot before, during and after irradiation. Yellow line represents the nuclear boundary. Scale bar: 5 μm .

(2) Fig. 1A, "H2B-GFP" should be "H2B-PAGFP".

Thanks for pointing out this typo, it has been fixed.

(3) Page 4, line 7, "While, at short time scales ($t < 5 \text{ min}$)..." Should " $t < 5 \text{ min}$ " be changed to " $t < 5 \text{ s}$ "?

Thanks for pointing out this typo, it has been fixed.

(4) Fig. 3C. Chromatin motion was further increased in ARH3 KO +PARGi cells under non-irradiation. Why? And what happens in ARH3 KO +PARGi cells under irradiation?

Previous reports have shown that cells lacking the ARH3 hydrolase display a persistent mono-ADP-ribose (MAR) signal on histones (PMID: 34019811). Due to this priming by a first ADPr moiety, treating ARH3 KO cells with PARG inhibitor leads to a spontaneous elongation of these marks, generating a strong poly-ADP-ribose (PAR) signal on the chromatin (Figure 3A and PMID: 34019811). Therefore, in these cells, the basal activity of PARP is sufficient to induce a strong ADPr signal in the absence of exogenous damage. We used this to assess the impact of MAR and PAR signals on histone dynamics in the absence of damage induction. The comparison of histone dynamics in wild-type cells and ARH3 KO treated or not with PARGi (Figure 3C) suggests that histone MARylation is already sufficient to promote histone mobility, which increases further upon PARylation. It seems reasonable to propose that these negatively charged ADPr polymers grafted on the nucleosomes promote a more dynamic chromatin conformation compared to the MAR marks. Histone mobility was even higher 1 min after damage in the ARH3 KO cells treated with PARGi (Figure R4), in line with the fact that DNA damage induction probably increased PAR signal further due to robust PARP1 activation at the lesions.

Figure R4: Mean jum-distance distributions of H2B in ARH3KO cells treated with PARGi (25 µM PDD00017273 for 24 hs) before and 1 min after laser irradiation at 355 nm. Tracks were obtained from 9 cells.

(5) Page 10, Line 3. "At higher folding scales, nucleosome dynamics was proposed as..."
A new modeling data is in a new preprint:

<https://doi.org/10.1101/2024.10.20.618801>

This reference has been added in the revised manuscript.

Reviewer #2 (Remarks to the Author):

In this manuscript, Fernandez and colleagues use single-nucleosome imaging combined with microirradiation to reveal that ADP-ribosylation (ADPr) of histones at DNA lesions induces transient chromatin decondensation, facilitating DNA repair. By observing chromatin at multiple scales, the authors show that PAR and MAR ADPr signals play distinct roles in initiating and maintaining chromatin decondensation. PAR deposition decondenses chromatin and increases nucleosome mobility, while MAR removal by ARH3 leads to chromatin recondensation. The findings highlight ADPr as a critical regulator of chromatin dynamics during the DNA damage response, enhancing our understanding of chromatin motion at DNA lesions.

While diffraction-limited imaging has been commonly applied to measure chromatin decondensation in response to DNA damage, to my knowledge, the application of single nucleosome tracking together with this assay is new. Although I am not an expert in the DNA damage response and thus cannot comment on the novelty or significance of the findings regarding the temporal destabilization/restabilization of nucleosome motion relative to chromatin condensation, to me the results were interesting. However, I feel that the manuscript could benefit from additional technical validation, particularly with respect to the neural network classifier and consistency in the histograms of the single-molecule jump distances prior to publication.

Major comments:

- The histograms of H2B jump distances appear very odd to me with multiple peaks that suggest separate sub-populations. However, I suspect that since these peaks seem to vary considerably across experiments and the histograms are presented as a continuous rather than binned distribution, it may just be that the histogram bins are too small to capture the consistent trends in the data. Please specify the bin size and indicate what guidelines were used in choosing the bin width. Please also include some metric of cellular and experimental variability in the datasets. While the authors list the number of cells measured for each condition, it's not clear if these came from independent experiments and all the data appear to be aggregated into the histogram. For clarity, an experiment (or cell-by-cell) breakdown of the histograms could be presented in the supplementary materials.

As suggested by the previous referee, we compared the mean jump-distance distributions obtained from different experiments and concluded that the three peaks observed on Figure 1C, which differs by only few nanometers, originate from day-to-day variability rather than subpopulations of histones within the immobile category analyzed on this graph (Figure R1). In the course of the revision process, we also revisited our pipeline to analyze the histone tracks and included a bootstrapping step, in line with the vbSPT approach previously described to analyze single-particle tracking data (PMID:

23396281). With this revised strategy, the distribution before damage induction displays a low variance (see revised Figure 1C).

Regarding the concern about histogram bins, we realized that the term histogram used in the figure legends of the original manuscript was inappropriate. Indeed, the mean jump-distance distributions presented on the different main figures are not shown as histograms but rather as kernel density estimation (KDE) curves, which are not affected by the choice of a specific bin. This use of KDE instead of histograms is now clearly stated in the revised figure legends. The parameters used in HTC are listed in the supplementary table.

- The implementation of a convolutional neural network to classify trajectories requires additional validation and documentation. It's unclear what the classifier was specifically trained on—were the linked trajectories used to generate synthetic images, which were then annotated into groups? Could the authors clarify this process as well as the overall network architecture?

We have included a more detailed description of the architecture and training approach of our CNN model in the revised method section of the manuscript (see p 16-17). Our overall strategy with the development of HTC was to be able to classify each trajectory as a whole. To reach this goal, each trajectory was converted into a trajectory-image prior to classification. We generated trajectory-images of 512x512 pixels to consider the sub-pixel accuracy of localization, assuming ballistic motion between two consecutive timepoints. Additionally, each segment along the trajectory-image was color-coded based on the instantaneous diffusion coefficient. Given that CNNs have been reported to be the most efficient approach for the classification of images (<https://doi.org/10.1145/3065386>; <https://doi.org/10.1162/neco.1989.1.4.541>), we used this approach to classify the generated trajectory-images. The overall network architecture was as follows: 512x512 size of input followed by 5 convolutional layers (C1: 8x8 - 32, C2: 5x5 - 64, C3: 2x2 - 128, C4: 2x2 - 256, C5: 2x2 - 512) where the corresponding numbers are the kernel size and the number of filters for each convolutional layer. Each convolutional layer was followed by max pooling and batch normalization layers with ReLu activation. The total number of trainable parameters is 762K. The optimization is performed with Adam on the cross-entropy loss where the learning rate and batch size are 0.0005 and 16 respectively. For the 3-classification, the softmax function is used at the end.

To train and validate the classifier, we manually annotated 1040 immobile, 1040 mobile and 1040 hybrid traces, which are now provided as supplementary data. These annotated trajectories (i.e. coordinates) were augmented 23 times with a rotation matrix $M(\theta) = \begin{bmatrix} \cos(\theta) & -\sin(\theta) \\ \sin(\theta) & \cos(\theta) \end{bmatrix}$ by increasing θ with $24/\pi$ radians for each rotation. The reason we augmented the trajectories at coordinate-level instead of at image-level is to avoid the distortion which happens inevitably when rotating pixelated images.

Additionally, I am uncertain why the authors chose this methodology over more established approaches, such as vbSPT (PMID: 23396281), which are commonly used

for classifying trajectories or Spot-On (29300163) to determine the fraction of molecules in different diffusive states. Classifying molecules transitioning between states has conventionally been an intractable task for shorter trajectories due to the probabilistic nature of diffusion. It seems that the authors could have attained most of the biological conclusions without developing this new classifier, but if they wish to include it, it needs additional validation and comparison to the other more commonly used methods referenced above. I'd also be curious whether the other more established methods mentioned above confirm the findings from the new CNN.

As explained above in our response to the first concern of referee 1, we also analyzed our data with vbSPT and compared the output with the one of HTC (see revised supplementary figure 2I). In contrast to our HTC approach, which considers each trajectory as a whole, vbSPT classifies individual segments within each trajectory. To compare the two methods, it was then necessary to allocate each trajectory to a given class based on the segment classification derived from the Viterbi path in vbSPT. Given that segments along the trajectories often switched classes, due to real changes in protein mobility but also to trajectory mislinking or localization errors, defining objective criteria was not straightforward. Therefore, we decided to define as immobile a trajectory whose all segments belonged to this class according to vbSPT output. With these restrictive settings, the immobile class defined by vbSPT showed lower mobility compared to HTC. Yet, both approaches report a dramatic increase in histone mobility upon damage. This shows that our HTC approach is able to detect mobility shifts similar to previously published methods such as vbSPT. It also highlights that this increase in histone mobility upon DNA damage is a robust phenotype that can be picked with various SPT analysis tools.

Of note, vbSPT classification is based on the estimation of the diffusion coefficient of molecules, with the assumption that they follow Brownian motion. The assumption of Brownian motion in vbSPT limits the generality of the program. For this study, histones, especially the slow population, are thought to be bound to DNA and thus to follow anomalous motion characterized by negative correlations between successive timepoints in contrast to Brownian dynamics (PMID: 28794266, 24998931). With the improvement of dyes in the last few years, we obtain trajectories of few tens of positions, thus approximating their motion with Brownian motion becomes less appropriate. Since HTC does not make specific assumptions regarding the diffusion regime, it might be more appropriate to analyze subdiffusive behaviors such as those expected for histones. To further support these claims, we now provide additional validation of our CNN-based approach for histone trajectory on the following link: [https://github.com/JunwooParkSaribu/HTC/blob/main/validation/Validation_process.ipynb]. This additional validation of CNN was performed on simulated fractional Brownian motion trajectories mimicking histone trajectories of lengths varying from 8 to 64. The validation result of HTC shows more than 99% accuracy for the simulated immobile histone trajectories regardless of their length. This is consistent with the result of HTC for the validation set of manually annotated data, which shows again the effectiveness of HTC for the classification histone trajectories and the reliability of our findings.

- The statement “While PARPi treatment slightly impacts the dynamics of the lacO array and the nucleosome in the absence of damage” seems to conflict with the trends in the

data provided. I agree that PARPi has minimal effects on the LacI in absence of irradiation. However, the effects on nucleosome motion in the PARPi condition are considerable. This can be seen in comparing blue distributions in Fig2A vs. Fig2B and those in FigS2C where PARPi treatment seems to cause a fairly dramatic increase in nucleosome mobility even without irradiation. In the GFP-LacI PARPi+/IR+ experiments (Supplementary Figure 2), there is a clear decrease in locus dynamics upon induction of DNA damage. For the nucleosomes PARPi+/IR+ experiments (Figure 2), there is an increase in steady state dynamics (comparing the blue "before" distributions in panels A, B, and C) and a loss of the transient increase in dynamics compared upon induction of DNA damage (comparing the red "1 min" distributions in panels A, B, and C). While I understand that the locus tracking and single-nucleosome imaging approaches do not measure exactly the same thing, it would be helpful for the authors to acknowledge and interpret these observed differences. To me, the data suggests that PARPi leads to a dramatic stabilization of lacI locus dynamics within 1 min after DNA damage while not affecting steady state locus dynamics. For nucleosomes, PARPi seems to destabilize nucleosomes at steady state, but prevent the additional increase in nucleosome dynamics after DNA damage. I'm unclear how these results fit into the model proposed in the paper.

Thank you for raising this unexpected finding. We confirm that, in the absence of damage, PARPi treatment leads to a significant increase in the motions of both the nucleosomes and the chromatin fiber (revised supplementary figure 4B). Besides break repair, PARP1 and ADP-ribose signaling were shown to regulate other cellular processes such as transcription and replication (see PMID: 35271815 for a recent review). We cannot exclude that an impact of PARPi on these functions may indirectly affect chromatin mobility at the fiber and nucleosome scales. PARP1 is able to dynamically associate with undamaged chromatin but the structural features of this interaction appear different from the complex formed between PARP1 and DNA breaks (see e.g. PMID: 31028139, 34919819). In particular, seminal work by the Kraus lab (PMID: 15607977) had proposed that PARP1 competes with H1 for binding to linker DNA, thus potentially affecting the chromatin compaction state. The activity displayed by PARP1 in this undamaged context seems quite different from the one at DNA breaks. Indeed, while the main residues modified by ADPr are Ser in the DNA damage context, targeting Glu and Asp residues on histones is central to ADPr-dependent regulation of transcription (PMID: 32822587). Given these differences, it appears conceivable that PARPi treatment does not show the same impact on PARP-dependent chromatin regulation in the damaged and undamaged contexts. We discuss these differences in the revised version of the manuscript (see the discussion, p 10).

Minor comments:

- The description of the chromatin compaction state measurement methods is unclear. It would be helpful for the authors to provide more detail on how the thickness of the photo-converted H2B line was estimated? Specifically, did they employ manual annotation, fit a Gaussian or another model, or use a different approach? Further elaboration on this step would greatly improve reproducibility.

To monitor the changes in the chromatin compaction state at DNA lesions, we thresholded the photoconverted H2B line using the Otsu algorithm and fitted the mask with an ellipsoid. The length of the small axis was used to estimate the thickness of the photoconverted line. This additional information is provided in the revised version of the methods section.

- In general, the authors employ somewhat imprecise terminology when describing chromatin. For example, there are numerous occasions wherein the authors utilize the term “chromatin folding” when referencing their results. For me, this is confusing. While I agree that the increase in the activated stipe width after DNA damage could suggest a chromatin decondensation, I’d hesitate to say anything about chromatin “folding” which suggests some change in how DNA wraps around nucleosomes or in which nucleosomes assemble into some higher-order structure. For me, this isn’t supported by the type of measurement performed in the data.

To clarify this point we changed the term “chromatin folding” to chromatin reorganization when referring to multiscale changes in chromatin compaction and dynamics. When referring specifically to the assay measuring the changes in the chromatin compaction state at DNA lesions, we now use the terms chromatin decondensation or loosening in the results and for the figure axes.

I’m also confused by the usage of the phrase ‘chromatin fiber’. The authors occasionally refer to mesoscale chromatin as ‘chromatin fiber,’ such as in the passage, ‘...most likely corresponds to H2B proteins stably incorporated into nucleosomes as they display an effective diffusion coefficient similar to that of the chromatin fiber.’ However, in other instances, they use ‘chromatin fiber’ to refer to individual nucleosomes, as in: ‘While the expression of wild-type PARP1 or PARP1-3SA in PARP1 KO cells both rescued the transient increase in nucleosome mobility at sites of DNA lesions, this was not the case for PARP1-LW/AA (Figure 4). These data demonstrate that it is the ADPr of histone and not PARP1 that triggers the increase in chromatin fiber mobility at sites of DNA breaks.’

Indeed, chromatin fiber refers exclusively to a mesoscale. We edited the text to avoid the inaccurate use of this term.

Additionally, in the discussion the authors state “persistent MAR signal could be sufficient to maintain an open conformation in the vicinity of the DNA lesions”. The term, “open conformation” is unclear here. I believe the authors are referring to the role in persistent MAR in preventing the recondensation of chromatin as indicated by shrinking in the width of the photoactivated line after DNA damage. To me, this suggests that MAR maintains a decondensed state rather than an “open conformation” which I would associate more with something like MNase accessibility. There may be other studies that relate histone ADPr to DNA accessibility (e.g. via MNase seq), but the data in this manuscript doesn’t measure “open-ness” vs. “closed-ness”. I feel like it’s important to be precise and consistent in the language used to describe specific results.

In a previous work, we showed that ADP-ribose dependent chromatin relaxation at DNA lesions is associated with increased DNase accessibility (PMID: 31566235). This increased accessibility of the DNA was also demonstrated by monitoring the recruitment of DNA binding domains such as the BZIP domain from C/EBPa to DNA lesions. In the revised version of the current manuscript, we show that the accumulation and release kinetics of BZIP closely matches those of the chromatin decondensation and recondensation phase (see our response to the fourth comment of referee 1 as well as the new supplementary figure 1B). Altogether, these data support the idea that the decondensed chromatin conformation observed at DNA lesions is associated with increased DNA accessibility.

- In the results, the authors state that “In contrast, chromatin remains in this decompacted state for several minutes despite the rapid drop in nucleosome dynamics”. I’m curious what biological process is responsible for stabilizing the nucleosome motion in the decondensed chromatin state? It might be worth revisiting this finding in the discussion section.

We showed previously that the histone deacetylase HDAC1 gets recruited to DNA lesions via its interaction with the chromatin remodeler CHD7 and that its deacetylase activity regulates the chromatin recondensation step (PMID: 33188175). Therefore, it is possible that HDAC1 also promotes the dampening of nucleosome motions by erasing acetylation marks on histones. Given that this dampening is impaired in the ARH3 KO cells (see Figure 5C), it would be interesting to investigate whether HDAC1 recruitment or activity at DNA lesions is affected by the loss of ARH3. These hypotheses are described in the revised version of the discussion.

- In the methods section for classification of the H2B tracks, the authors refer to a “convoluted neural network (CNN)” I believe that this should be “convolutional neural network”.

Thanks for pointing out this typo, it has been fixed.

Reviewer #4 (Remarks to the Author):

The authors present an interesting new take on understanding changes to chromatin at sites of DNA damage. Moreover, they show how this movement is regulated by waves of PARylation and MARYlation. Together, their study establishes a new model how ADP-ribosylation controls nucleosome mobility, triggering a transient breathing of chromatin, crucial for initiating the DNA damage response. This study is timely and will be of great interest to the scientists working in the genome stability and chromatin fields of research.

Specific comments:

In figure 1A the authors use 405 nm irradiation to induce DNA damage, which clearly induces large scale chromatin relaxation. In figure 1B and C however, the authors switch to using 355 nm irradiation. Do the authors see the same relaxation kinetic with 355 nm laser as with 405 nm laser?

To assess chromatin decondensation after irradiation at 355 nm, we first tried to use the same assay as with the irradiation at 405 nm, based on the simultaneous induction of DNA lesions and photoactivation of fluorescently tagged H2B. Unfortunately, the intensity of the pulsed 355 nm laser required to induce DNA lesions was bleaching the photoactivatable fluorophores that we tested (both PAGFP and PATagRFP), thus preventing an efficient fluorescent tagging of the damaged chromatin. Consequently, we modified our assay and rather monitored the thickness bleached line generated upon irradiation at 355 nm in cells expressing H2B-EGFP as now shown on revised supplementary figure 1A. This analysis demonstrated a rapid decondensation of the chromatin after irradiation at 355 nm, similar to what was observed with the 405 nm laser. We also found that PARPi treatment did not only inhibit this chromatin decondensation process but induced an overcompaction, in line with our previous observations upon 405 nm irradiation (see PMID: 31566235). Altogether, these findings demonstrate that both irradiation methods, either with a continuous 405 nm illuminating Hoechst-presensitized cells or with a pulsed 355 nm laser without presensitization, both induce similar ADP-ribose dependent chromatin decondensation at DNA lesions.

Do you see any discernible difference in histone movement in the middle of your irradiated line as compared to the edges of the stripe? Or do you see equal movement of histones across the whole stripe? Are there areas of chromatin compaction at the edges of stripes?

We tried to analyze nucleosome motions in subregions of the irradiated line to identify potential differences in chromatin dynamics in the middle and at the edge of the damaged area. This approach reduced the number of tracks available in each subregion to build the mean jump-distance distributions, thus affecting their accuracy. Therefore, we were unable to detect differences in dynamics between histones located in the middle or at the edge of the irradiated line using the current data. Nevertheless, we also made use of our assay based on the phototagging of the damaged chromatin to compare changes in the compaction state in the middle and at the edge of the irradiated line. When performing irradiation with a 405 nm laser in cells expressing H2B fused to the photoconvertible dye Dendra2, we observed a different tagging pattern compared to cells expressing H2B-PAGFP as shown on Figure 1A. Instead of tagging the irradiated line, we were able to label its edges, probably because the strong laser intensity at the middle of the irradiated line was bleaching the photoconverted Dendra2. With this pattern, we were able to monitor the thickness of the photoconverted lines (averaged between the two lines) as well as the distance between these lines (see Figure R5 below). While this distance gave access to chromatin decompaction in the damaged region, the thickness of the lines could be used to assess the compaction state at the border of this region. Using this assay, we found that in contrast to the middle of the damaged region that undergoes a rapid decondensation immediately after damage induction, the edges of this

area initially displays a slight compaction, as shown by a reduction in the thickness of the photoconverted lines, followed by a delayed decompaction. These findings suggest that chromatin decompaction is initiated from the middle of the damaged region, thus pushing away nearby chromatin leading to its transient compaction, and then progressively transmits to the periphery of the damaged area. These data are quite interesting and would require further investigation to better understand the underlying process but we believe that such detailed analysis goes beyond the scope of the current manuscript. Therefore, we prefer not to include these data in the revised version of our work.

Figure R5: **A.** U2OS cells expressing H2B-Dendra2 were irradiated with a 405 nm laser after sensitization with Hoechst. This generated a photoconversion pattern composed of two lines tagging the edges of the irradiated area. The temporal evolution of this pattern could be used to assess changes in chromatin compaction in the middle and at the edges of the damaged area. **B.** On images as shown in A., the average thickness (T_h) of the lines as well as their distance (D) could be monitored as a function of time after irradiation. The curves correspond to the mean \pm SEM of 13 cells.

Did you compare chromatin movement in damaged cells treated with PARG inhibitors?

We analyzed histone motions in WT cells treated with PARGi and observed an increased mobility 1 minute after irradiation (see Figure R6 below) which is however less pronounced than in the untreated cells (compare with Figure 1C). Then, histone motions dampen 10 min after damage as in the untreated condition. We showed previously that PARGi treatment impairs the rapid erasing of PAR signal at DNA lesions while leaving the overall kinetics of MAR marks unaffected (PMID: 37116497). Therefore, our data suggest that it is not the rapid loss of PAR signal that triggers the drop of histone mobility following its initial increase upon damage. Given that we found increased histone mobility at 10 min after damage in ARH3 KO cells that display impaired removal of the MAR signal (Figure 5C), the dampening of histone dynamics might rather be controlled by MARYlation. Nevertheless, in untreated WT cells, this dampening seems to occur as early as 5 min after damage, i.e. at a time when MAR signal is at maximum. Therefore,

altogether, these data do not currently provide a clear mechanism to explain the rapid decline of histone dynamics few minutes after damage. As proposed above in our response to one comment from reviewer 2, this process may involve histone deacetylation by HDAC1 but a more thorough analysis going beyond the scope of the current manuscript would be needed for a better description of this process. Yet, the fact that an active mechanism might be needed to dampen histone motions and, ultimately, to recondense the chromatin, is in line with our hypothesis that ADPr signaling promotes a switch between two stable chromatin conformations, a condensed and a looser one, upon DNA damage.

Figure R6: H2B motions in PARGi-treated WT cells upon DNA damage. Mean jump-distance distributions of H2B in WT cells treated with PARGi (25 μM PDD00017273 for 24 hs) before, 1 and 10 min after laser irradiation at 355 nm. Number of analyzed cells: $N_{\text{bef}}=11$, $N_{1\text{min}}=11$, $N_{10\text{min}}=5$.

Have the authors had the opportunity to compare chromatin dynamics or potential chromatin disassembly with over expression of chromatin remodellers – for example, what happens with the PAR-dependent chromatin remodeler ALC1?

We showed in previous publications (PMID: 27733626 and 37106138), that the rapid chromatin decondensation observed at DNA lesions relies on both ATP-dependent and -independent mechanisms. More specifically, we also reported that several remodelers such as CHD1L/ALC1 or CHD7 contribute to the relaxation process via their ATP-dependent nucleosome remodeling activity (PMID: 27733626 and 33188175). The multiscale analysis pipeline that we established in the current work would indeed be well suitable to assess in more detail the exact contribution of each of these remodelers, in particular regarding nucleosome dynamics and stability. Nevertheless, we believe that such detailed analysis goes beyond the scope of the current manuscript and should rather be presented in a future manuscript.

Introduction. 'While PARG is the most active PAR hydrolase, ARH3 is a specific serine MAR eraser.' It would be worth also specifying here that the serine-linked ADP-ribosylation is the most robust form of ADP-ribosylation in DNA damage response and that the histones are the major targets (doi: 10.7554/eLife.34334).

We edited the introduction as follows: "Upon recruitment to DNA lesions, the polymerase PARP1 adds ADP-ribose on the serine residues of nearby proteins, primarily PARP1 itself and histones."

POINT BY POINT RESPONSE:

Point by point response to the comments of the referees regarding the manuscript by García Fernández et al.

We thank the referees for their positive feedback regarding the revised version of our work and for supporting publication provided that we can address the remaining concerns raised by referee #2. We revised further the text of the manuscript to address these concerns and we edited the typos pointed out by the referees. Our point-by-point response to the specific concerns can be found below in blue.

Reviewer #1 (Remarks to the Author):

The authors addressed my comments with additional data. I would support this revised paper for publication.

Page 18, Line 13. "1minut" -> "1 minute".

Thanks for pointing out this typo, it has been fixed.

Reviewer #2 (Remarks to the Author):

In this revised manuscript, Fernandez and colleagues have addressed many of my (and the other reviewer's comments). I commend the authors for including additional explanation and discussion regarding the neural network classifier and the display of the single molecule tracking data. Overall, I feel that the manuscript is a useful contribution to the field with interesting and careful experimental results. I believe it is likely suitable for publication in Nature Communications provided that the authors can address my remaining concerns detailed below.

1) I still have concerns about how the single molecule tracking data is presented throughout the manuscript, which is very unusual compared to most single molecule tracking/imaging experiments in most of the literature. Specifically, in the revised manuscript, the authors have clarified how the CNN was trained by using the trajectories to generate artificial "images" which can then be classified using convolutional filters. This is an interesting approach and could be useful for future studies. However, my primary remaining concern is that the network was trained on "manually annotated" examples of real molecule trajectories. I suppose for clear cases like H2B where the slow diffusing state is substantially different from the free diffusing state, this may still be fairly accurate. And the fact that vbSPT provides similar results is assuring. However, it does mean that the CNN is essentially being trained to match a subjective classification that may vary from user to user and likely would not generalize well for settings where molecular motion can be more complex (e.g. 3-state diffusion for transcription factors that display both specific and non-specific binding) that cannot be easily classified by eye. Given that quantifying the number of molecules in different diffusive states is a relatively minor part of the manuscript, I would still be in favor of publication provided that the authors include a clear description of these limitations in the

manuscript discussion.

As pointed out by this referee, our CNN-based approach has been trained for the detection of slowly moving proteins that likely correspond to histones stably incorporated into nucleosomes within the chromatin fiber. We acknowledge that this analysis tool is not readily generalizable to the analysis of the motion of other factors, in particular if they display higher mobility compared to histones. To specifically state this limitation of our approach, we included the following paragraph in the revised discussion:

"For the analysis of nucleosome motions, we established a new CNN-based approach (HTC for Histones Trajectory Classifier) to classify the single trajectories and extract step distributions. While this approach has been optimized for the detection of nucleosomes stably incorporated into chromatin, applying it to study the dynamics of other factors would require a thorough retraining and validation of the classifier especially for short trajectories."

2) My other concern is how the jump-length distributions are presented in the revised manuscript. If I understand correctly based on the updated methods section, for most of the figures in the paper, the authors are plotting the kernel density estimator for a bootstrapped analysis on the mean jump distance after filtering for the stable population of trajectories using their CNN classifier. Is this correct? While it may still be "correct" to justify the authors claims, it is very different from how this data is usually displayed. It should be made clear in the main text that the authors are fitting the expected mean jump distance for just the stable population of H2B trajectories after filtering. The authors also need to provide details in the methods section for the fitting procedure used to generate the KDE (what software/function) from the filtered mean jump distances.

In the legends of the different figures including jump distance distributions, we specifically state that they correspond to KDE plots for the immobile population of H2B tracks inside the irradiated region. Furthermore, to state more clearly that our analysis focuses on the slowly mobile population of histones, we included the following sentences in the middle of page 5 in the results section:

"For the rest of this study, our analysis of histone motion focused on the population of slowly moving H2B proteins likely incorporated into the nucleosomes. Histone mobility was assessed by measuring the mean distance covered in 10 ms for each H2B trajectory, a generic metric that did not require to assume a specific diffusion model: the data are then presented as the probability density plot of bootstrap distribution, calculated using a kernel density estimation (KDE) (see Methods)."

Furthermore, we now included additional information in the methods section to better describe the approach used to generate the KDE plots:

"We used the SciPy package of Python (Virtanen et al., 2020) to obtain the bootstrap distributions and the PDF (probability density function) plots with kernel

density estimation using Scott's rule (Scott, 1992). All the averaged mean jump distances were equally weighted in KDE."

Minor comments:

1) First paragraph of the "Results" section the authors state: "U2OS cells expressing H2B fused to the photo-activatable dyes PAGFP". PA-GFP is a protein, not a dye. Please also spell out the acronym upon first usage in the text.

This sentence has been edited and now reads: "U2OS cells expressing H2B fused to the photoactivatable green fluorescent protein (PAGFP)"

2) The green MSD curve at 10 min after IR in Figure 1B seems to show an upward "kink" at 10 seconds. Based on the error bars, this is remarkably repeatable between experiments. Can the authors comment on or explain this phenomenon?

We thank this referee for the careful examination of our data. We went back to the original tracks used to build the mean MSD curve for the condition far from damage at 10 min post-irradiation. We noticed that among the tracks, one was showing a significant drift during part of the acquisition due to bead movement used for drift correction. The associated MSD curve showed a major upward curvature that appeared as an outlier compared to the other individual MSDs but was sufficient to distort the mean MSD curve. Therefore, we cut the fraction of the track showing the drift and recomputed the averaged MSD curve that is now shown on the updated figure 1B. We believe that this new curve is more representative to the average behavior of the lacO array away from the irradiated region.

3) Supplementary Figure 3: The color coding in Figure panel A is switched between the trajectory map and the plot to the right.

The color coding has been corrected for this panel.

4) Supplementary Figure 4A: Please clarify how PARP1 recruitment was measured. The legend indicates WT U2OS cells, but I'm assuming the cells were either transfected or stably expressing some form of fluorescently labeled PARP1?

We edited the legend of this figure to provide additional information regarding the cell sample as well as the quantification of PARP1 recruitment:

"(A) Kinetics of PARP1 recruitment to DNA damage induced by 355 nm irradiation in PARP1 KO U2OS cells transiently transfected with EGFP-PARP1, left untreated or treated with 30 μ M of Talazoparib (N=10 for each condition). Recruitment was estimated by measuring the mean fluorescence intensity within the manually segmented irradiated region. This fluorescence signal was background subtracted and corrected for photobleaching by dividing it to the mean intensity of the whole nucleus. This corrected intensity was then normalized to the value prior to damage."

5) Figure 2A is missing a callout in the main text.

Figure 2A is now referenced in the text (see middle of page 6)

Reviewer #4 (Remarks to the Author):

The authors have fully and thoroughly addressed my comments.